



# Enhanced Markov-Type Categorical Prediction with Geophysical Soft Constraints for Hydrostratigraphic Modeling

Liming Guo[1], Thomas Hermans[1], Nicolas Benoit[3], David Dudal[2], Ellen Van De Vijver[1], Rasmus Madsen[4], Jesper Nørgaard[4], and Wouter Deleersnyder[1,2]

[1]Department of Geology, Ghent University, Krijgslaan 281-S8, 9000 Ghent, Belgium
[2]Department of Physics, KU Leuven Campus Kortrijk–Kulak, Etienne Sabbelaan 53 bus 7657, 8500 Kortrijk, Belgium
[3]Geological Survey of Canada (Natural Resources Canada), Quebec City, Canada
[4]Geological Survey of Denmark and Greenland (GEUS), 8000 Aarhus, Denmark

**\* Corresponding author:** Liming Guo (liming.guo@ugent.be)

**Abstract.** Accurately characterizing hydrostratigraphic structures is essential for reliable groundwater flow and transport modeling. Due to limited borehole coverage and geological complexity, uncertainty analysis plays a vital role in supporting robust hydrogeological modeling. Traditional geostatistical approaches such as Multiple-Point Statistics (MPS), offer flexibility in reproducing complex geological patterns and uncertainties, but they are computationally demanding, may struggle to maintain stratigraphic consistency, and can be difficult to apply in practice. Alternatively, the Markov-type Categorical Prediction (MCP) framework has a better computational efficiency and enforces stratigraphic ordering. However, its effectiveness is challenged in areas with sparse borehole data. To address this limitation, this study presents an enhanced MCP approach that incorporates airborne electromagnetic (AEM) geophysical data as soft probabilistic constraints on lithology occurrence. A tunable parameter controls the relative contribution of geophysical and geological information, allowing for flexible data integration within the simulation process. The approach is tested on both synthetic and real-world cases. Synthetic experiments of different scenarios demonstrate that incorporating geophysical constraints improves lithological prediction accuracy, particularly when combined with borehole data. In the field application from Egebjerg, Denmark, we demonstrate how a statistical relationship between lithology and resistivity can be derived by integrating SkyTEM data with borehole lithological logs and depth information. That relation is then combined with conditional probabilities from training images extracted from a 3D interpreted model, using the MCP framework. The results show that the integrated approach enhances the generations of complex geological features, such as buried valleys, especially in areas with limited direct observations. By embedding geophysical information into the MCP framework, the method combines the spatial consistency and stratigraphic ordering of MCP with the extensive coverage and subsurface sensitivity of geophysical data. This integration overcomes a key limitation of MCP and enables more reliable simulations in regions where direct subsurface observations are limited, providing a practical and adaptable tool for improving geological modeling in groundwater studies.



## 1 Introduction

In hydrogeological modeling, uncertainty analysis is essential to account for the incomplete knowledge of subsurface conditions and to support robust decision-making in water resource management, contamination risk assessment, and infrastructure planning (Harken et al., 2019; Previati et al., 2025; Feyen and Caers, 2006; Hermans et al., 2023). Subsurface environments are inherently heterogeneous and data available for modeling, such as borehole logs, are typically sparse and unevenly distributed in space. Without explicitly quantifying uncertainty, predictions based on deterministic models risk to misrepresent hydrogeological processes and may lead to ineffective or even damaging management strategies (Koch et al., 2014; Vilhelmsen et al., 2019). This is particularly true when deterministic models are used as the basis for groundwater flow and solute transport simulations, where small inaccuracies in hydrostratigraphic structure can lead to significant deviations in predicted flow paths, travel times, and contaminant breakthrough curves (Dai et al., 2017; Zuo et al., 2023; Song et al., 2019). By incorporating uncertainty into the simulation of geological heterogeneity, geostatistical approaches provide not only plausible geological scenarios but also essential input for ensemble-based hydrogeological forecasting, which is one type of probabilistic approach that relies on multiple realizations to assess model uncertainty. (Moore et al., 2022; Zimmerman et al., 1998; Enemark et al., 2024; Hermans et al., 2015).

In constructing geostatistical models, borehole data are the most commonly used conditioning data because they provide direct observations of subsurface lithology (Boyd et al., 2019; Madsen et al., 2021). However, the cost, time, and logistical constraints of drilling often result in insufficient borehole coverage, especially at greater depths or in poorly accessible regions (Bongajum et al., 2013; Madsen et al., 2022). In contrast, geophysical exploration methods, such as electrical resistivity tomography (ERT) or airborne transient electromagnetic (AEM) surveys, offer a cost-effective way to obtain relatively high-density spatial coverage (Maurya et al., 2023; Steuer et al., 2020; Prikhodko et al., 2024; Védrine et al., 2023). Although geophysical data do not directly measure lithology, they provide property contrasts (e.g., in resistivity) that, after inversion and interpretation, can be statistically linked to hydrofacies distributions (Michel et al., 2020; Looms et al., 2008).

Within geostatistical modeling, constructing a reliable hydrostratigraphic model typically requires integrating multiple sources of information, including hard data (e.g. borehole observations), probabilistic constraints (e.g. resistivity models derived from inverted airborne geophysical measurements), and prior geological knowledge about spatial variability and continuity. (He et al., 2014; Høyer et al., 2017; Barfod et al., 2018). Each of these sources contributes complementary information: geophysical data provide indirect insights into large-scale hydrostratigraphic architecture, boreholes offer detailed but localized information, and TIs capture prior geological knowledge and spatial continuity. Combining them leads to more robust and geologically realistic models (Barfod et al., 2018; Jha et al., 2014). However, this integration is not straightforward, as it requires a quantitative understanding of the relationships between different data types. Without careful treatment, combining data of varying resolution, quality, and interpretability may lead to overinterpretation or underutilization of valuable information (Levy





et al., 2024).

Multiple-Point Statistics (MPS) has become a widely used geostatistical method in hydrostratigraphical modelling. MPS uses Training Images (TIs) to quantify the spatial variability and reproduce complex geological patterns that cannot be captured by traditional two-point geostatistics (Mariethoz et al., 2010). In recent years, several studies have explored introducing probabilis-
tic spatial constraints, including geophysical models, into MPS frameworks. For example, Barfod et al. (2018) demonstrated that conditioning MPS simulations with airborne electromagnetic data significantly improved the delineation of buried valleys in Denmark. Madsen et al. (2021) proposed treating uncertain geological interpretations as probabilistic constraints, comparing MPS and Gaussian simulation methods, and showed that MPS produced more geologically plausible and connected realizations. Hermans et al. (2015) developed a full MPS-based inversion framework that used ERT data both to falsify prior
geological scenarios and to locally constrain groundwater simulations, showing the strength of MPS in quantifying uncertainty and integrating multiple data types. Despite these advances, several challenges remain, MPS methods are computationally expensive and highly sensitive to training images that are geologically realistic and representative of the entire site (He et al., 2013; Levy et al., 2024). A training image of limited extent can lead to biased simulations that misrepresent key structural features. Moreover, the integration of geophysical data is often heuristic and lacks a formally probabilistic structure (Meerschman
et al., 2013). Many existing implementations of geostatistical simulation, such as MPS-based methods, cannot enforce explicit geological transition rules (e.g., forbidding clay above limestone), which means that they may fail to represent stratigraphic relationships accurately, particularly those relevant to spatial shifts in facies proportions or variations in layer thickness. Both limitations are critical when simulating subsurface structures for groundwater flow and transport modeling (Cordua et al., 2016; Kim et al., 2017).


A recently applied geostatistical approach by Benoit et al. (2018), known as Markov-type Categorical Prediction (MCP), provides an alternative framework to traditional multiple-point statistics (MPS) for simulating categorical geological units. MCP uses bivariate transition probabilities derived from a training image. One of the key advantages of MCP is that it reduces the dependence on high-quality or highly repetitive training images, which can be a limiting factor in some MPS implementations
(Allard et al., 2011). When key features in the TI are sparse, irregular, or unique, MPS may struggle to reproduce them consistently, potentially leading to artificial discontinuities or oversimplified realizations (Barfod et al., 2018). By contrast, MCP operates on a different principle. Rather than trying to reproduce entire patterns from the TI, MCP uses pairwise transition probabilities between units to capture the likelihood of one unit being adjacent to another (Benoit et al., 2018). This approach allows MCP to extract essential geological information in a non-stationary fashion without needing a complete TI. Further-
more, MCP remains computationally efficient—even when simulating models with a large number of lithological categories because it avoids high-order pattern scanning or search-tree construction. One of MCP's strengths is its ability to strictly respect geological rules when certain transitions between units are geologically impossible. For example, if a specific lithological unit is never observed directly above another in the training data, MCP ensures that this configuration will not appear in the simulated model based on zero bivariate probability of these two units (Benoit et al., 2018). Yet, previous applications of the





MCP framework have relied almost exclusively on hard conditioning data, such as borehole lithology. In settings where such
data are sparse, the method often defaults to random simulation, which can result in geologically unrealistic outputs (Benoit
et al., 2018). However, MCP offers greater transparency and flexibility in conditioning, making it well suited for the integration
of soft information derived from geophysical inversion models. To leverage this potential, our study extends the MCP frame-
work by incorporating geophysical soft constraints into the simulation process. This integration aims to reduce uncertainty and
enhance the geological realism of subsurface models, particularly in areas that are poorly constrained by hard data.

This paper is organized as follows. Section 2 introduces the principles of the Markov-type Categorical Prediction (MCP)
framework and outlines its extension for integrating geophysical soft constraints. Section 3 presents the results in two parts. In
Section 3.1, a synthetic test case is developed, where a true lithological model with multiple layers is defined and conductivity
values are assigned using a Gaussian random field. Airborne electromagnetic data are simulated via forward modeling and sub-
sequently inverted using a 1D inversion scheme. The resulting inverted conductivity model is then statistically linked to the true
lithology to derive a stochastic relationship, which is incorporated into the MCP framework as an additional constraint. Section
3.2 applies the proposed approach to a real-case study in Egebjerg, Denmark. Bivariate transition probabilities are computed
from multiple 2D transects extracted from a 3D interpreted geological model and merged using the Extended Logistic Opin-
ion Pool (ELOP) method to ensure directional consistency. The resulting representative transition probabilities serve as priors
for MCP simulations along all transects, enabling the construction of a coherent 3D lithological model. Two representative
transects, located in areas with shallow boreholes, are selected for detailed analysis to evaluate the added value of geophysi-
cal constraints. Borehole lithology, borehole depth information, and inverted resistivity from airborne electromagnetic (AEM)
data are statistically linked, and this relationship is combined with MCP probabilities through a tunable integration param-
eter. Finally, constrained MCP simulations are performed to generate multiple lithological realizations, allowing uncertainty
quantification and assessment of geophysical constraints. The results are discussed in Section 4.

## 2   Methodology

### 2.1   Markov-Type Categorical Prediction

Markov-Type Categorical Prediction (MCP) is a probabilistic approach to generate categorical geological models that effec-
tively maintains spatial relationships in a computationally efficient fashion. Within the MCP framework, it is assumed that
the influence of neighboring categories on the target location can be captured through individual transition probabilities from
the target to each neighbor, with independent interactions among the neighbors themselves. This conditional independence
assumption simplifies computations by considering that the categorical states of neighboring points are independent once the
category at the target location is known. Although a simplification, it has been shown to produce accurate and unbiased predic-
tions in some geostatistical applications (Benoit et al., 2018).



Given a set of known categories at neighborhood locations $x_1, x_2, \ldots, x_n$, the conditional probability of category $i_0$ at an unsampled location $x_0$ is calculated as (Allard et al., 2011):

$$P_{i_0|i_1,\ldots,i_n}^{\mathrm{MCP}} = \frac{p_{i_0} \prod\limits_{k=1}^{n} p_{i_k|i_0}}{\sum\limits_{i_0=1}^{I} p_{i_0} \prod\limits_{k=1}^{n} p_{i_k|i_0}} = \frac{p_{i_0}^{1-n} \prod\limits_{k=1}^{n} p_{i_k,i_0}}{\sum\limits_{i_0=1}^{I} p_{i_0}^{1-n} \prod\limits_{k=1}^{n} p_{i_k,i_0}}, \tag{1}$$

where $i_0$ is the category being predicted, and $i_k$ are the observed categories in the neighborhood of $x_0$. The bivariate probabilities $p_{i_k,i_0}$ quantify the likelihood of co-existence between two categorical states. The marginal probability $p_{i_0}$ represents the prior likelihood of category $i_0$; in this study, it is defined as the mean proportion of each lithology observed in the training image.

A key step in MCP implementation involves defining the spatial range of the search neighborhood ($x_n$) around each estimation point. This range is a user-defined parameter that can be tuned to balance spatial continuity and computational cost. Following Benoit et al. (2018), we adopted an octant-based search strategy with at least one neighbor per octant, resulting in up to eight conditioning data per location.

The MCP approach requires a representative Training Image (TI), from which the bivariate probabilities $P(i_k, i_0)$ are derived and used in Equation (1). These probabilities describe the likelihood of observing a pair of categories $(i, j)$ separated by a specific lag $h$, and they are computed directly from the TI. To compute the bivariate probability between categories i at location $x$ and category $j$ at location $x + h$, we use the cross-indicator covariance:

$$P(I(x), J(x+h)) = E[I(x)J(x+h)], \tag{2}$$

where $I(x)$ and $J(x+h)$ are binary indicator functions defined as $I(x) = 1$ if the category at location $x$ is $i$, and 0 otherwise. Similarly, $J(x+h)$ takes value one if j is observed at point $x + h$. The expectation is taken over all valid pairs of grid nodes separated by vector $h$ in the TI. The expectation is computed over all points with separation vector $h$. The estimation of the probabilities is efficiently handled by a method by Marcotte (1996), which relies on the fast Fourier transform to calculate correlations accross multiple spatial lags.

The method is implemented using a sequential simulation framework, where categorical values are assigned iteratively based on the computed conditional probabilities using Equation (1). The simulation begins with a blank grid, except for available hard data (e.g., borehole observations) . At each iteration, a location $x_i$ is selected following a random path. If sufficient neighboring points are found within the defined octant-based search range, the conditional probability for each possible category at that location is computed using the bivariate transition probabilities. If no neighbors are available, the simulation falls back to using marginal probabilities alone derived from the training image. Once the conditional probability distribution is determined, a category is sampled using random sampling, and the simulated value is assigned to the location. The newly simulated point



then is added to the conditioning neighborhood and contributes to the estimation of subsequent nearby nodes. This process continues until the entire simulation domain is completed, resulting in a single categorical realization. The procedure can be

repeated multiple times to generate an ensemble of realizations, which can then be used for uncertainty analysis.

A distinctive feature of MCP is its zero-forcing property, which guarantees that if a bivariate probability between two categories is zero, the corresponding category transition over a considered distance is completely restricted in the generated realizations (Allard et al., 2011). This property makes MCP particularly suitable for modeling stratigraphic sequences where ordering con-

straints must be preserved (Benoit et al., 2018). However, in regions with limited conditioning data, MCP may revert to using marginal probabilities when neighborhood information is insufficient, which can lead to the occurrence of geologically incompatible transitions. To address this, a post-processing correction procedure is introduced in Section 2.3 to enforce geological consistency in the final simulations.

## 2.2 Integration of geophysical data

Geophysical data can provide additional constraints to geostatistical simulations by linking lithological categories with physical properties (Guo et al., 2024). In this study, a stochastic resistivity-lithology relationship is established by deriving conditional probabilities from inverted resistivity models. Unlike most previous MPS based approaches, which typically assume ideal geophysical sensitivity or overlook its degradation with depth, our method explicitly incorporates both the smoothing effects

of regularized inversion and the loss of resolution with depth into the probabilistic framework (Hermans and Irving, 2017; Hermans et al., 2015; Barfod et al., 2018; Deleersnyder et al., 2023). These probabilities represent the likelihood of observing specific lithological classes given the resistivity values at corresponding locations.

To integrate this information with the categorical constraints from the MCP framework, the permanence-of-ratios principle

is applied. This principle assumes that the relative influence of different data sources can be preserved through a ratio-based formulation. The intermediate quantities used in this formulation are defined as follows (Journel, 2002; Isunza Manrique et al., 2023):

$$a = \frac{1 - P(A)}{P(A)}, \quad b = \frac{1 - P(A|B)}{P(A|B)}, \quad c = \frac{1 - P(A|C)}{P(A|C)}, \quad x = \frac{1 - P(A|B,C)}{P(A|B,C)}, \tag{3}$$

where $P(A)$ represents the marginal probability of lithology $A$, $P(A|B)$ is the conditional probability of lithology $A$ based on

MCP constraints $B$, and $P(A|C)$ reflects the conditional probability of lithology $A$ given the geophysical data $C$. The final joint probability $P(A|B,C)$, conditioned on both sources, is obtained using the following proportionality:

$$\frac{x}{b} = \left(\frac{c}{a}\right)^{\tau}. \tag{4}$$




Solving for $P(A|B,C)$ yields:

$$P(A|B,C) = \frac{a^\tau}{a^\tau + b \cdot c^\tau}. \tag{5}$$

Here, the exponent parameter $\tau$ controls how strongly the model emphasizes differences between the components $a$ and $c$. As $\tau$ increases, the contrast between these terms becomes more pronounced, allowing the model to more clearly favor the source (prior or geophysical) with the stronger signal. This enables flexible adjustment of the relative influence of geophysical data, depending on the level of confidence in its relationship to the target property.

Once the joint probabilities are computed for all categories and locations, multiple realizations of the subsurface lithology are generated using the sequential simulation framework of the MCP algorithm. The integration of resistivity based probabilistic information serves to guide the categorical simulation toward geologically plausible outcomes that also honor the spatial patterns suggested by the geophysical data.

**2.3 Post-processing**

Although the MCP framework has the property to preserve stratigraphic ordering through directional transition probabilities, violations may still occur in practice. Such inconsistencies can arise during the early stages of simulation, when no neighboring points are yet available and the algorithm must rely solely on marginal probabilities. Besides, in scenarios where borehole (hard) data are sparse, larger weight is assigned to the geophysical soft constraints, thus soft constraints derived from geophysical data

may inadvertently steer the simulation toward geologically unrealistic category choices. To ensure geological plausibility and spatial continuity in the final realizations, two rules of thumb-based correction procedures are implemented to identify and resimulate inconsistent cells.

1. Neighborhood consistency check: Each simulated cell was compared with its 24 current neighbors within a $5 \times 5$ window. If fewer than 9 neighbors had the same lithological category as the center cell, it is marked as inconsistent and is
resimulated again.

2. Vertical age-consistency check: This rule enforces stratigraphic order by checking the relative age of lithological units. For each cell, up to 6 cells directly above it are evaluated. If at least one of those upper neighbors represented older lithologies (had a higher category number), the current cell was flagged as violating geological logic and marked for resimulation.

Once inconsistent cells are identified through the rules above, they were removed and resimulated again through framework. Then, all successfully simulated values from previous iterations are retained and treated as hard conditioning data in the sequential simulation approach. This means that as the correction process progresses, the MCP framework becomes increasingly guided by a growing set of accepted values, which helps to stabilize the simulation and maintain spatial consistency. The pro-



cess is repeated until either the number of incorrect cells remained unchanged for three iterations or a maximum of 40 iterations

was reached. At that point, the final simulation result is presented.

## 3    Results

### 3.1    Synthetic case

To validate the integration of geophysical constraints into the Markov-type Categorical Prediction (MCP) framework, a synthetic case study was designed. This experiment aimed to assess the ability of the integrated approach to accurately reproduce

lithological structures under different data availability conditions. By integrating geophysical data into the MCP simulation process, we evaluate the method's capacity to reduce uncertainty and improve geological realism prior to real-world application.

The complete workflow of the synthetic case is summarized below and visually illustrated in Figure 1:

1. A synthetic three-layer lithological model was constructed on a 2D $80 \times 50$ grid to represent the true subsurface structure.

2. A Gaussian random field was used to generate spatially correlated conductivity values within each lithology, based on the mean and standard deviation values in Table 1.

3. Synthetic AEM data were generated from the conductivity model from Step 2 with the 1D TDEM forward modelling routine available in th SimPEG package (Heagy et al., 2020).

4. The synthetic AEM data were then inverted using a 1D deterministic inversion. The inversion employed a smoothness-
constrained regularization scheme, generating inverted conductivity model.

5. The bivariate probabilities are derived from the a synthetic training image, which are then used to calculate the conditional probabilities for the MCP function as in Equation (1). In the constrained scenarios, a single borehole located X=8m is introduced as hard conditioning data.

6. A stochastic resistivity-lithology relationship was derived by linking the inverted conductivity values to the true lithology
classes at each grid cell (Hermans and Irving, 2017), which were integrated into the MCP framework following the approach outined in Section 2.2.




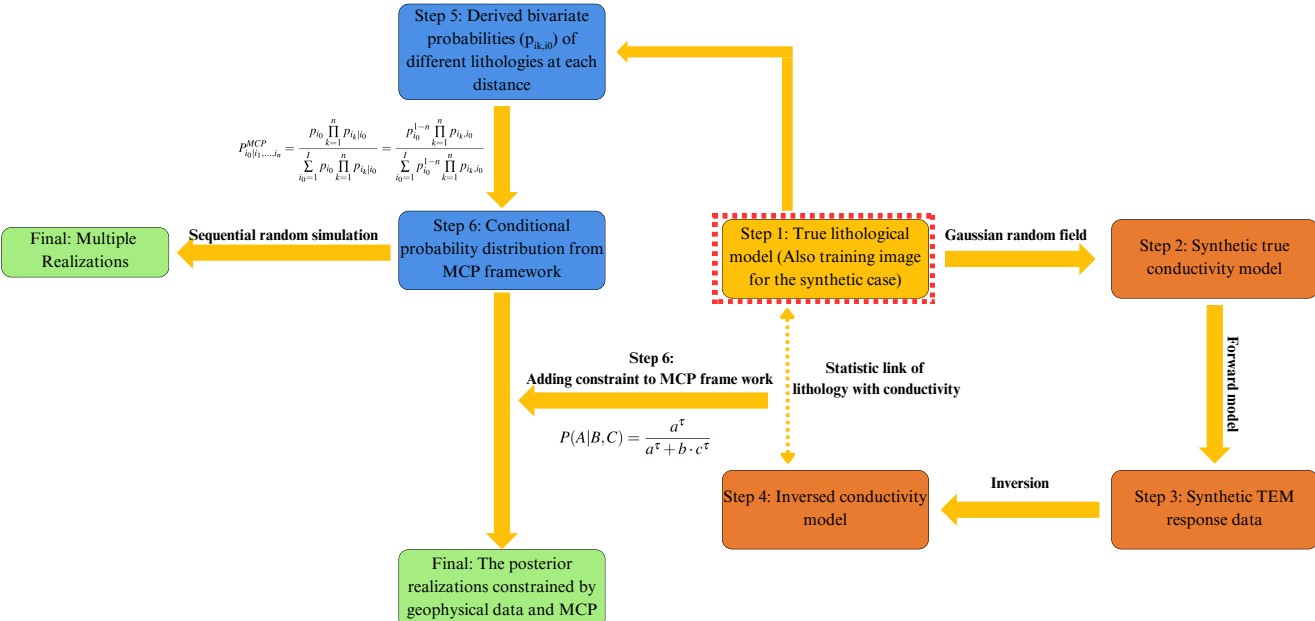

**Figure 1.** Workflow for the synthetic case

### 3.1.1 Model Setup

The synthetic lithological model consists of three distinct lithological units, as shown in Figure 2 (left). The associated electrical conductivity model (right) is generated by assigning values to each lithology using a Gaussian random distribution, which corresponds to an isotropic Gaussian covariance model, with the correlation length controlled by the standard deviation of the filter. The specific parameters used for each lithology including mean, standard deviation, and correlation length are summarized in Table 1.

**Table 1.** Conductivity parameters assigned to each lithological layer in the synthetic model.

| Layer | Mean Conductivity (S/m) | Standard Deviation | Correlation length |
|-------|-------------------------|--------------------|--------------------|
| 1 | 0.15 | 0.06 | 3 |
| 2 | 0.75 | 0.15 | 3 |
| 3 | 0.35 | 0.06 | 4 |





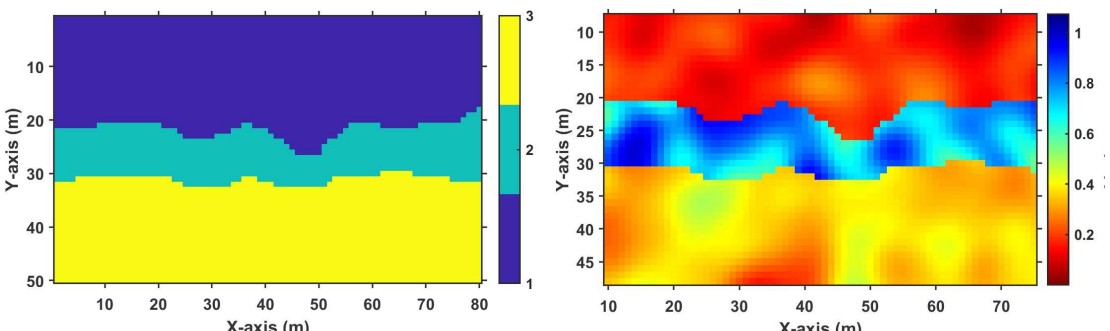

**Figure 2.** Left: Synthetic lithological model. Right: Synthetic conductivity distribution generated from the Gaussian random field.

### 3.1.2 Geophysical Forward and Inversion Modeling

The forward data were generated with a 1D time-domain forward modelling routine based on synthetic lithological model,
mimicking a circular loop source at 20 meters height, with a step-off waveform and 21 logarithmically spaced time channels
ranging from $10^{-5}$ to $10^{-2}$ seconds. 5% multiplicative Gaussian noise was added to the synthetic data. We discretized a voxel-
type inversion model using 30 layers with increasing thicknesses, ranging from 0.5 m to 5 m, to mimic the decreasing vertical
resolution of airborne TEM surveys. The layer thicknesses follow a $\log_{10}$ spacing with depth to reflect the logarithmic decay of
resolution in geophysical measurements. The inverse problem was solved using a projected Gauss-Newton conjugate gradient
(GNCG) optimizer, with Tikhonov-type regularization (Constable et al., 1987) applied to stabilize the solution. Figure 3 shows
the synthetic data and inversion models.

These inverted profiles are then statistically linked to the true lithology by evaluating the joint occurrence frequencies be-
tween conductivity and lithology classes across the grid. From this, we compute the conditional probability of lithology given
conductivity, denoted as $P(\text{Lithology} \mid \text{Conductivity})$, which is used as the soft geophysical constraint in the MCP framework:

$$P(\text{Lithology} \mid \text{Conductivity}) = \frac{P(\text{Lithology}, \text{Conductivity})}{P(\text{Conductivity})}, \tag{6}$$

where $P(\text{Lithology}, \text{Conductivity})$ is the joint probability distribution obtained from co-occurrence counts in the synthetic grid,
and $P(\text{Conductivity})$ is the marginal probability of conductivity.





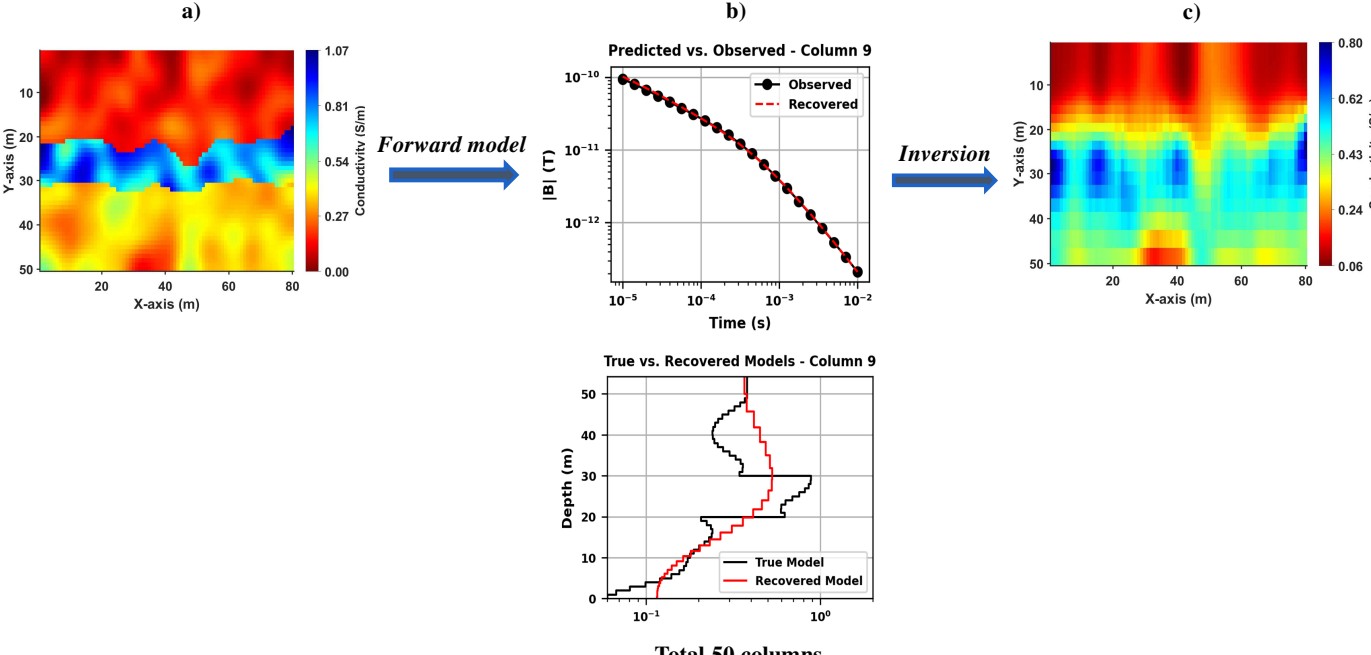

**Total 50 columns**

**Figure 3.** (a) True synthetic conductivity model constructed over 50 vertical soundings (columns) using random Gaussian distribution. (b) Both observed and fitted TEM responses for example of Column 9, and comparison between the true and inverted conductivity profiles (bottom). (c) Final stitched 2D conductivity section obtained by combining 1D inversion results for all columns.

### 3.1.3 Four simulation Scenarios

To evaluate the influence of different conditioning data types on the MCP framework, we designed four simulation scenarios that incorporate (or not) borehole data and geophysical constraints in varying combinations (Table 2), which isolates the respective and combined impacts of boreholes and inverted geophysical data on lithological prediction.





**Table 2.** Simulation scenarios evaluated in the synthetic case.

| Scenario | Description |
|---|---|
| Unconstrained MCP Simulation (S1) | No external constraints; only training image is used. |
| MCP with Borehole Constraints (S2) | A single borehole located at X = 8m is included as a hard data constraint. |
| MCP with Geophysical Constraints (S3) | Inverted conductivity used as soft probabilistic constraint. |
| MCP with Both Constraints (S4) | Both borehole and geophysical constraints applied simultaneously. |

For each scenario, 100 realizations were generated to assess the variability and consistency of the predicted lithological structures.

### 3.1.4 Synthetic Results

To evaluate how different constraints affect simulation results, we focus on the probability of predicting lithology category 2 across all realizations. In synthetic case, no post-process step is applied. Layer 2 was selected because it represents the central unit in this model, offering an indirect view on overlying and underlying layers. The results of the four scenarios are summarized in Figure 4, which shows three realizations from each scenario, along with the probability distribution of lithology 2 based on 100 simulations.

In the first scenario, no conditioning data were applied and the simulations were entirely driven by the spatial patterns from the training image. This led to considerable variability between realizations and a poor representation of the true model. Adding a single borehole in the second scenario significantly improved the simulation near the borehole. The model showed higher certainty in the vicinity of the hard data, and the predicted lithologies more closely matched the true values at those points. However, further away from the borehole, uncertainty remained high. The third scenario, which only introduced geophysical constraints, resulted in better large-scale continuity of the lithological units. Since the inverted resistivity model provides spatially distributed information, the simulation was able to capture broader geological trends, even though some details between the second and third layer remained somewhat uncertain, which reflects the loss of resolution with depth for TEM data. In the final scenario, where both borehole and geophysical constraints were used, the simulation achieved the best overall performance. The combination of point-based hard data and spatially distributed soft information led to more accurate and geologically consistent results. The realizations were more stable across the entire domain, and the predicted lithological patterns closely followed the structure of the synthetic true model.



In conclusion, The synthetic results from Figure 4 demonstrate the complementary roles of hard and soft conditioning data. Borehole constraints effectively suppress simulation artifacts and enforce stratigraphic consistency, whereas geophysical constraints improve structural resolution, particularly in the delineation of transitional layers such as layer 2. However, when applied in isolation, soft constraints may lead to overconfident yet geologically implausible realizations. This emphasizes the need to integrate both data types within the MCP framework to achieve simulations that are both structurally accurate and geologically realistic. Moreover, although MCP has the property of enforcing hydrostratigraphical ordering, the incorporation of soft constraints from geophysical data may still result in local violations of this ordering. Therefore, in the following real-field application, a post-processing step, as described in Section 2.3, is necessary to ensure geological plausibility.



**Figure 4.** Example realizations (top three rows) and probability distribution of lithology 2 (bottom row) based on 100 simulations under four constraint scenarios (no post-process step for synthetic case).





### 3.1.5 Sensitivity analysis of the geophysical constraint

As demonstrated in the previous results, realizations generated using the MCP framework alone strictly adhere to geological
ordering constraints. However, in regions lacking hard conditioning data, the simulations tend to revert to more uniform or
overly smoothed patterns. In contrast, realizations influenced solely by geophysically derived conditional probabilities exhibit
greater local variability but often lack spatial coherence. To explore this trade-off, a sensitivity analysis for the parameter $\tau$
(which governs the relative influence of geophysical data in Equation (5)) was conducted.


Introducing the Jaccard dissimilarity index, $J(S,R)$:

$$J(S,R) = \frac{|S \cap R|}{|S \cup R|}, \quad U(S,R) = 1 - J(S,R), \quad \overline{U(S,R)} = \frac{\sum\limits_{i=1}^{n} U(S,R_i)}{n}, \tag{7}$$

Which quantifies the overlap between the true model dataset $S$ and a single realization $R$. From this, we define the correspond-
ing Jaccard dissimilarity, $U(S,R) = 1 - J(S,R)$, which reflects the degree of mismatch between two categorical maps. To

assess the overall variability across multiple realizations, we compute the averaged Jaccard dissimilarity, $\overline{U(S,R)}$, as the mean
dissimilarity over $n$ realizations.

Figure 5 presents the Jaccard dissimilarity for the different importance of geophysical data introduced, reinforcing the effec-
tiveness of moderate geophysical weighting. As shown in Figure 5, the averaged Jaccard dissimilarity decreases as $\tau$ increases

to a certain point, indicating an improved agreement between the simulations and the true model. The minimum dissimilarity
is achieved at $\tau \approx 3$, beyond which further increases in $\tau$ do not significantly improve model accuracy and may even lead
to a little bit over-reliance on potentially noisy geophysical data. However, once $\tau$ becomes sufficiently large, the influence
of varying $\tau$ values on the simulation results stabilizes. These results indicate that assigning a relatively high geophysical
weighting $\tau$ within the MCP framework can lead to improved agreement with the true lithological model. Although excessive

weighting may introduce minor inconsistencies, the overall performance remains good. This finding provides the guidance for
the following real-field applications in areas with sparse borehole data, where assigning a relatively higher geophysical weight
may effectively enhance model performance.





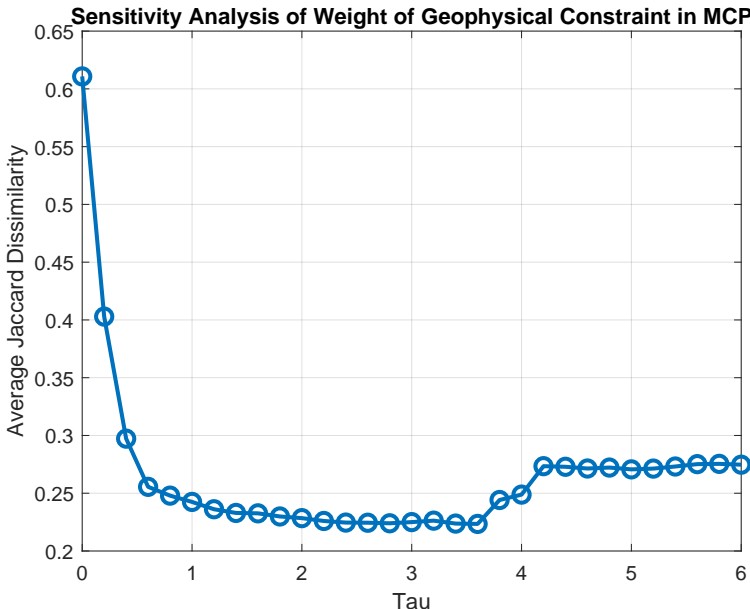

**Figure 5.** Sensitivity of weight of geophysical constraint in MCP via averaged Jaccard dissimilarity.

## 3.2 Real-case application

### 3.2.1 Study area

The study area is located near Egebjerg in Jylland, Denmark, and covers approximately 150 km$^2$ (Figure 6). It is situated in a glacially influenced landscape characterized by a complex subsurface geological setting, where buried valleys, discontinuous aquitards, and interbedded sand and clay units contribute to significant spatial heterogeneity. The surface geology is primarily composed of clayey tills, with terrain elevations reaching over 160 meters above sea level (Madsen et al., 2022). The study area has been interpreted using a stratigraphic model comprising 14 hydrostratigraphic units, ranging from Quaternary sediments to

Danian limestone (Table 3). This parameterization follows the standard stratigraphic framework adopted in the national Danish hydrostratigraphic model (Jørgensen et al., 2010a). In particular, Layers $5, 6, 7, 8$ and $9$ in Table 3, which consist of Quaternary sand and clay deposits, are commonly found within buried valley structures and are known to exhibit irregular thicknesses and discontinuities (Jørgensen et al., 2010b; Sandersen and Jørgensen, 2017). These features present challenges for the subsurface modeling due to their highly variable nature and the limited availability of deep conditioning data.


Borehole lithological information was obtained from the Danish national borehole database, Jupiter, which includes 794 boreholes within the study area (Hansen and Pjetursson, 2011). However, approximately 78% of these boreholes are shallower than 50 meters (Ter-Borch, 1991). To complement the limited borehole coverage, airborne transient electromagnetic (TEM) data



from the SkyTEM system were utilized. These data were acquired along flight lines spaced 150 to 250 meters apart, providing

dense spatial coverage across much of the area (Madsen et al., 2022; Sørensen and Auken, 2004).

**Table 3.** Lithological layers in the catchment with descriptions (Adapted from Madsen et al. (2022))

| Layer | Name | Description |
| --- | --- | --- |
| Layer 1 | Quaternary Sand 1 | Northwestern sand deposits from meltwater. |
| Layer 2 | Quaternary Clay 1 | Clay deposits mainly in the northern region. |
| Layer 3 | Quaternary Sand 2 | Sand and gravel layers in northern plateaus. |
| Layer 4 | Quaternary Clay 2 | Clay deposits in plateau areas. |
| Layer 5 | Quaternary Sand 3 | Sand-rich layers in shallow buried valleys. |
| Layer 6 | Quaternary Clay 3 | Clay sediments in valleys. |
| Layer 7 | Quaternary Sand 4 | Meltwater sand deposits in buried valleys. |
| Layer 8 | Quaternary Clay 4 | Clay-rich layers in valleys. |
| Layer 9 | Quaternary Sand 5 | Sand layers found in deep buried valleys. |
| Layer 10 | Miocene Clay 1 | Mica-rich clay from Miocene erosion. |
| Layer 11 | Miocene Sand 1 | Deltaic sand deposits from Miocene. |
| Layer 12 | Miocene Clay 2 | Deep Miocene clay with high mica content. |
| Layer 13 | Paleogene Clay | Regional Paleogene clay and marl deposits. |
| Layer 14 | Limestone | Cretaceous limestone underlying the area. |





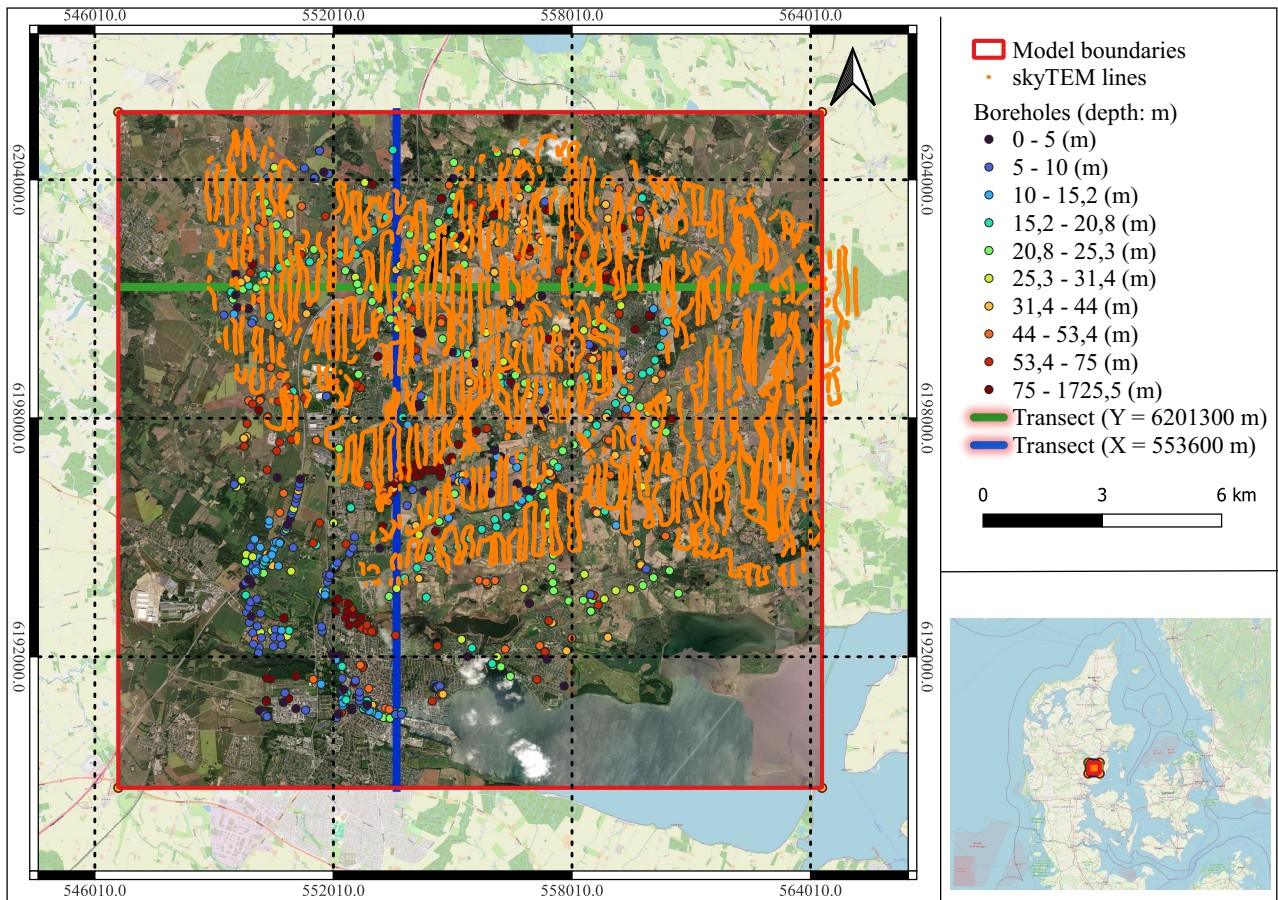

**Figure 6.** Distribution of boreholes and SkyTEM flight lines in the study area near Egebjerg, Denmark. The airborne electromagnetic data provide dense spatial coverage across the region in the central and northeastern parts. The red frame outlines the boundary of the previously established 3D model, which is used as a reference model for training image extraction. Borehole data are mostly shallow, with the majority of boreholes reaching depths of less than 50 meters. Two highlighted transects $X = 553600$ m and $Y = 6201300$ m were selected for detailed analysis. Both transects include segments with limited borehole data coverage.

### 3.2.2 Interpreted model and selection of training images

The 3D geological model of the study area was established by a cognitive hydrostratigraphic model, which integrates borehole data, geophysical surveys and geological interpretations (Enemark et al., 2022; Madsen et al., 2022). This model serves as a reference for training image generation and uncertainty assessment. This 3D geological model as shown in Figure 7 is originally a layer-based model without regular discretization in the vertical direction. In this study, we impose a uniform grid on the model domain, with horizontal extents from $X = 546600$ m to $564300$ m and $Y = 6188700$ m to $6205700$ m in 100 m intervals, and vertical discretization from elevation $Z = 80$ m to $-100$ m in 5 m intervals. Structurally, the hydrostratigraphic model




consists of 13 geological boundaries that separate six aquifers (primarily Quaternary and Miocene sands) from seven aquitards

(composed mainly of clays and marls). These layers presented at Table 3 extend from the top-limestone surface, which ranges

from -120m in the north, to -550 in the south, to the terrain surface (Jørgensen et al., 2010b; Ter-Borch, 1991). Figure 7 shows

the 3D interpreted model along with representative 2D transects. This model is used as the basis for extracting training images,

which are then used to constrain the MCP simulations and guide the generation of realistic geological realizations.

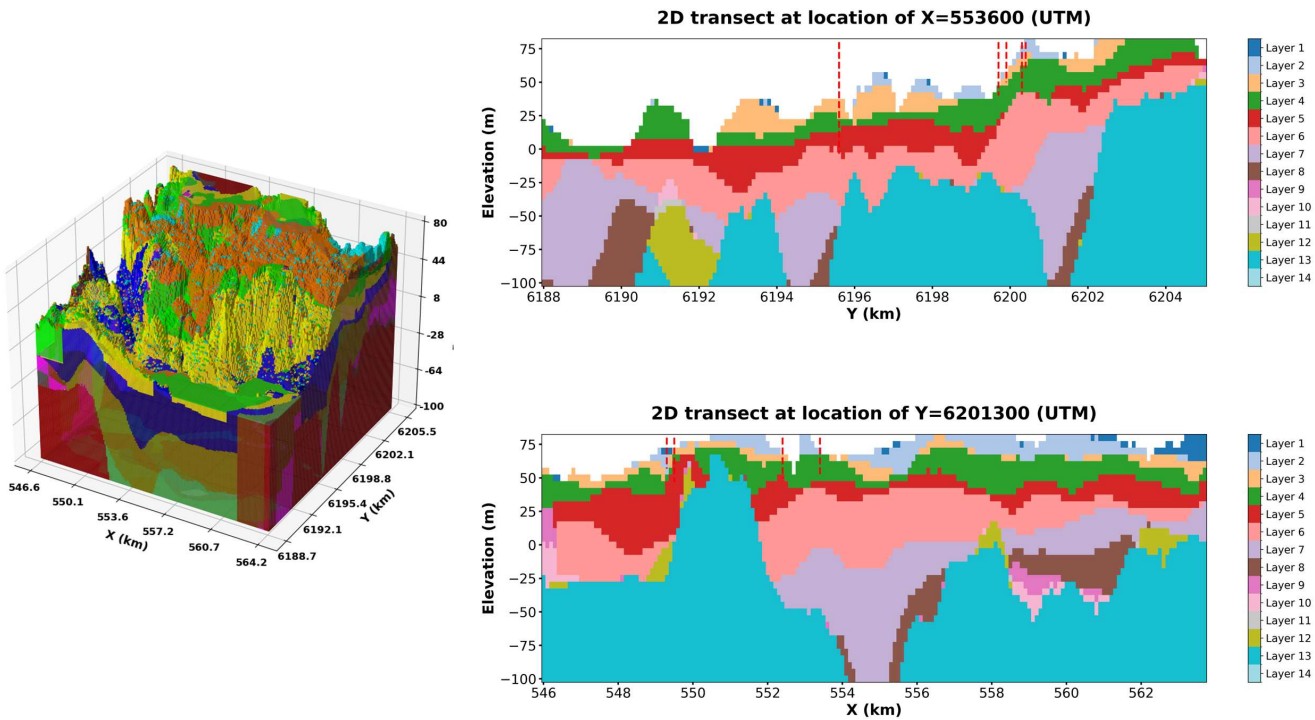

**Figure 7. Left:** Interpreted 3D geological model used as prior information for training image extraction.

**Right (Top):** Selected 2D transect at $X = 553600$ m used for detailed MCP simulation analysis.

**Right (Bottom):** Selected 2D transect at $Y = 6201300$ m used for detailed MCP simulation analysis.

Red dashed lines indicate borehole locations and their depth extent.

The training image (TI) plays a central role in the MCP framework, as it is used to derive the bivariate transition probabilities

$p_{i_k,i_0}$, which describe the spatial relationships between lithological categories. Since a complete 3D TI is often unavailable in

practice, We mimic this limitation by extracting 2D transects every 400 meters along both the X and Y directions from the

interpreted 3D model, resulting in 45 slices in the X-direction and 43 in the Y-direction. These 88 transects reflect the spatial



variability and layering observed in the prior information and are used to derive bivariate probabilities for the MCP algorithm.


To integrate the spatial information from the different individual transects into a single representative set of bivariate probabilities, we apply the Extended Logistic Opinion Pool (ELOP) method (Satopää et al., 2014). ELOP combines multiple probability estimates in a statistically consistent way, enabling the aggregation of directional spatial patterns while preserving variability.

$$\hat{p} = \frac{\prod_{i=1}^{N} p_i^{w_i}}{\prod_{i=1}^{N} p_i^{w_i} + \prod_{i=1}^{N} (1-p_i)^{w_i}}, \tag{8}$$

where $p_i$ represents the bivariate probability derived from each individual 2D transect, and $w\_i$ is the corresponding weight assigned to each bivariate probability, here equal weights are assigned to all transects.

The resulting unified 2D probability dataset is then used as input for MCP simulations across the entire model domain. Figure 8 shows an example of the integrated bivariate probability distribution $P(\text{Lithology} = 12, \text{Lithology} = 12)$, constructed from all

88 extracted transects using the ELOP function. Hence, the integrated bivariate probability serves as prior spatial information for all transects of the 3D model and enables the construction of a full 3D lithological simulation by applying the MCP framework across each transect sequentially.

Two two-dimensional transects were extracted from the interpreted 3D geological model: one aligned along $X = 553600$ m and

the other along $Y = 6201300$ m. These transects were selected in zones characterized by limited borehole coverage and variable SkyTEM data availability, which are typical of areas where conventional modeling approaches often struggle to represent complex geological features. The spatial variability along these transects, especially the presence of buried valleys and heterogeneous stratigraphic transitions, provides a interesting test case for the MCP framework. While MCP has demonstrated robust performance in synthetic cases with relatively uniform stratigraphy (Benoit et al., 2018), its effectiveness in capturing highly

variable geological features in real-world applications depends heavily on the quality and density of available constraints.





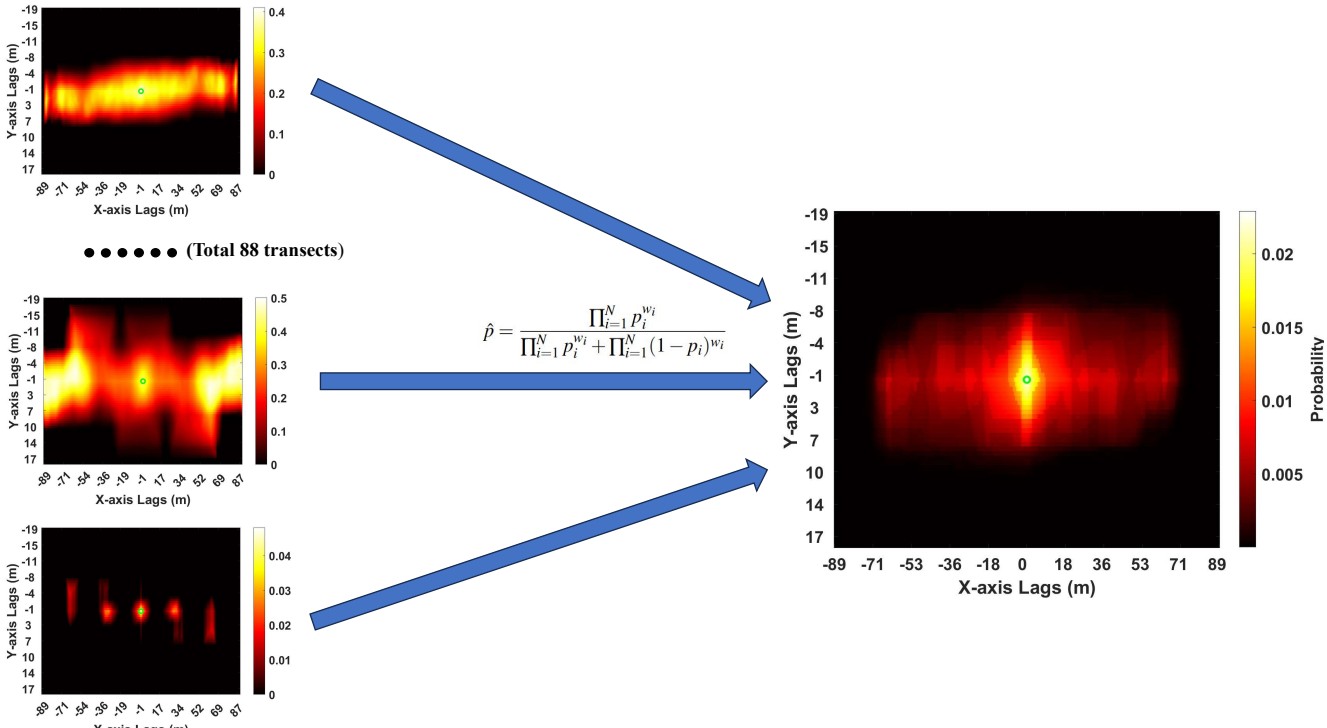

**Figure 8.** Bivariate transition probabilities $P(i, j)$, where $i$ and $j$ represent the 14 lithological layers, were derived from 88 transects extracted from the 3D interpreted model. These were then merged into a single generalized 2D bivariate probability distribution using the Extended Logistic Opinion Pool (ELOP) method. This unified distribution can be applied across all 2D transects. Shown here is an example for $P(\text{Lithology} = 12, 12)$ illustrating the spatial relationship aggregated from all 88 transects.

### 3.2.3 Geophysical data

For this case, we used the existing resistivity model from the SkyTEM system to characterize the subsurface structure, which is located north of the Horsens Fjord in Jutland, Denmark, covering approximately 300 km$^2$ (Madsen et al., 2022). The TEM data were collected along flight lines spaced between 150 and 250 meters, providing comprehensive spatial coverage. Data acquisi-
tion involved transmitting electromagnetic pulses into the ground, with the subsequent decay of induced currents measured to detect variations in electrical resistivity (Sørensen and Auken, 2004). Early-time decay measurements provided high-resolution information for shallow depths, while later-time measurements offered insights into deeper geological layers, enabling investigations down to depths exceeding 300 meters. SkyTEM is particularly well-suited for this application due to its high spatial coverage, its sensitivity to both shallow and deep conductive structures, and its ability to detect subtle resistivity contrasts in
layered sediments (Jørgensen et al., 2010b; Sandersen and Jørgensen, 2017). Early-time signals resolve near-surface features with high resolution, while late-time signals penetrate deeper, enabling characterization of complex features such as buried





valleys, aquifers, and aquitards down to depths exceeding 300 meters (Sørensen and Auken, 2004).

Unlike the synthetic case, where a 1D deterministic inversion was used, the TEM data were inverted using the Laterally Con-
strained Inversion (LCI) method (Auken et al., 2005), which applies spatial constraints to ensure lateral continuity between
adjacent soundings, resulting in more geologically consistent resistivity profiles (Madsen et al., 2022). This is especially impor-
tant in areas with limited borehole control, as it reduces inversion artifacts and noise (Høyer et al., 2015; Jørgensen et al., 2015).

To establish a statistical relationship between borehole information and the inverted resistivity model, resistivity values along
each borehole depth profile are first estimated by applying linear interpolation using measurements from nearby SkyTEM
flight lines (Figure 9). As shown in Figure 9c, the resistivity distributions associated with different lithological classes exhibit
significant overlap. This makes it challenging to distinguish between lithologies based on resistivity alone. To improve it, depth
is introduced as an additional conditioning dimension. To estimate conditional probabilities of lithology given resistivity, the
interpolated resistivity values are statistically linked with lithological classifications and depth values from the boreholes. This
is formulated as:

$$P(\text{lithology} \mid \text{depth}, \text{resistivity}) = \frac{P(\text{lithology}, \text{depth}, \text{resistivity})}{P(\text{depth}, \text{resistivity})} \tag{9}$$

However, since the study area contains only a few deep boreholes, direct sampling of conditional probabilities from sparse data
can introduce considerable noise and lead to unstable estimates. To address this, and to ensure a more robust and continuous
estimation of the borehole–resistivity relationship, Kernel Density Estimation (KDE) is employed in place of simple histogram
based sampling (Manrique et al., 2024). By modeling the joint probability distribution in a continuous probabilistic framework,
KDE not only helps smooth out local variability but is particularly well-suited to this scenario, where depths are only available
at sparse borehole locations and resistivity values are derived from constrained 1D inversions, thereby yielding more reliable
conditional probabilities in data-sparse zones. The resulting conditional probability distributions for each lithological unit are
shown in Figure 10.



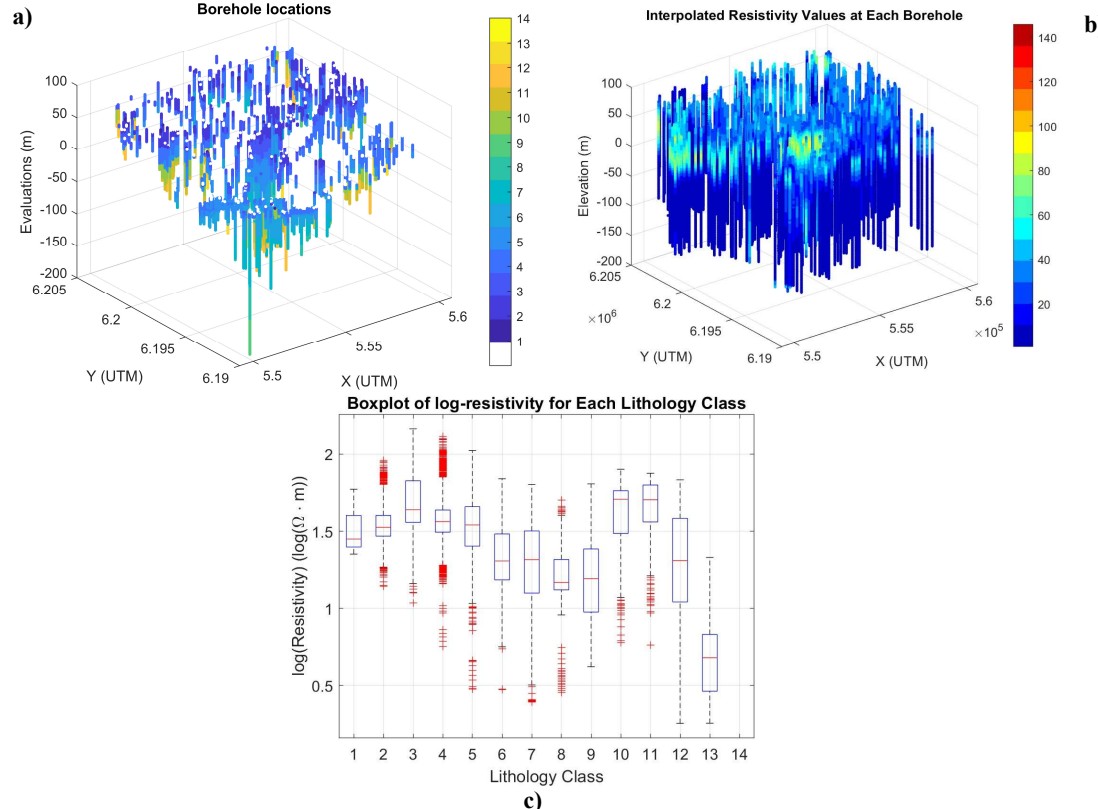

**Figure 9. (a)**: Lithological information of each borehole. **(b)**: Inverted resistivity values are interpolated along the borehole locations. **(c)**: Inverted resistivity distribution with each lithology type based on interpolated resistivity values.







**Figure 10.** The heatmaps present the smoothed probabilistic distributions of each lithological layer estimated via KDE, conditioned on resistivity and depth. The red lineplot in each subplot represents the global normalized marginal probability of lithology conditioned solely on resistivity, derived from the KDE results. The boxplots below each heatmap display the distribution of resistivity values at each layer based on inverted geophysical data at borehole locations.

### 3.2.4 Workflow for the field case

The modeling workflow for the real-case application is summarized and illustrated in Figure 11. The first step is to extract prior geological information from different transects of a 3D interpreted model, which are used to derive bivariate transition probabilities that represent spatial relationships between lithological units. The second step involves deriving a stochastic relationship that links borehole lithologies and their depths with resistivity values interpolated from the inverted model at borehole positions, using the neighboring TEM soundings. The final step is to integrate both geological expert knowledge




represented in the TI and geophysical information using the permanence of ratios principle Equation (5)) within the MCP framework, generating geologically realistic lithological realizations constrained by both borehole and geophysical data.

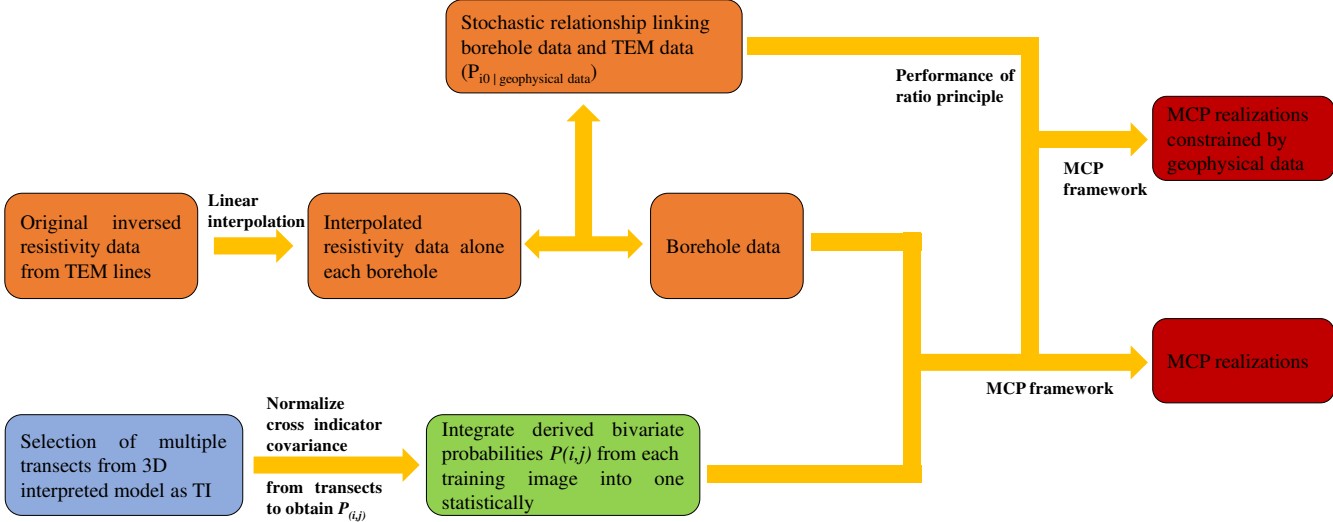

**Figure 11.** Workflow for realistic case.

### 3.2.5  Simulation results of real-case study

The simulation results for the selecting transects are presented for two scenarios: (i) using only the training image and borehole data, and (ii) incorporating additional geophysical constraints derived from interpolated resistivity profiles.

Figures 12 and 13 present two example realizations for each scenario at both two transects, more realizations can be seen in the Appendix. In the absence of geophysical constraints and with limited hard data (borehole) support, the MCP simulations tend

to produce more simplified and uniform stratigraphic configurations. This tendency is particularly evident in the reduced presence of buried valley features, typically associated with Layers 5 to 9. These valley layers are poorly delineated, with simulated transitions between lithologies appearing diffuse and inconsistent with the expected buried valley morphology. The associated expectation maps for two transects indicate the tendency of all simulations to produce overly uniform or vertically stacked layers in regions lacking hard conditioning data. Nevertheless, the MCP framework still effectively maintains the stratigraphic

ordering of lithological units, ensuring that the simulated realizations respect the geological age relationships encoded in the training image and conditioning data.





By contrast, when geophysical constraints are introduced (bottom of Figures 12 and 13), the simulations demonstrate markedly improved spatial realization and yield realizations that are more geologically consistent and closely match the existing refer-

ence model (Figures 7). Buried valleys, in particular, are more clearly defined, and the geometries of Layers 5–9 show better agreement with the interpreted stratigraphy. The expectation maps further confirm that the inclusion of AEM resistivity data enables sharper delineation of complex subsurface features and significantly improves lateral continuity across stratigraphic boundaries.

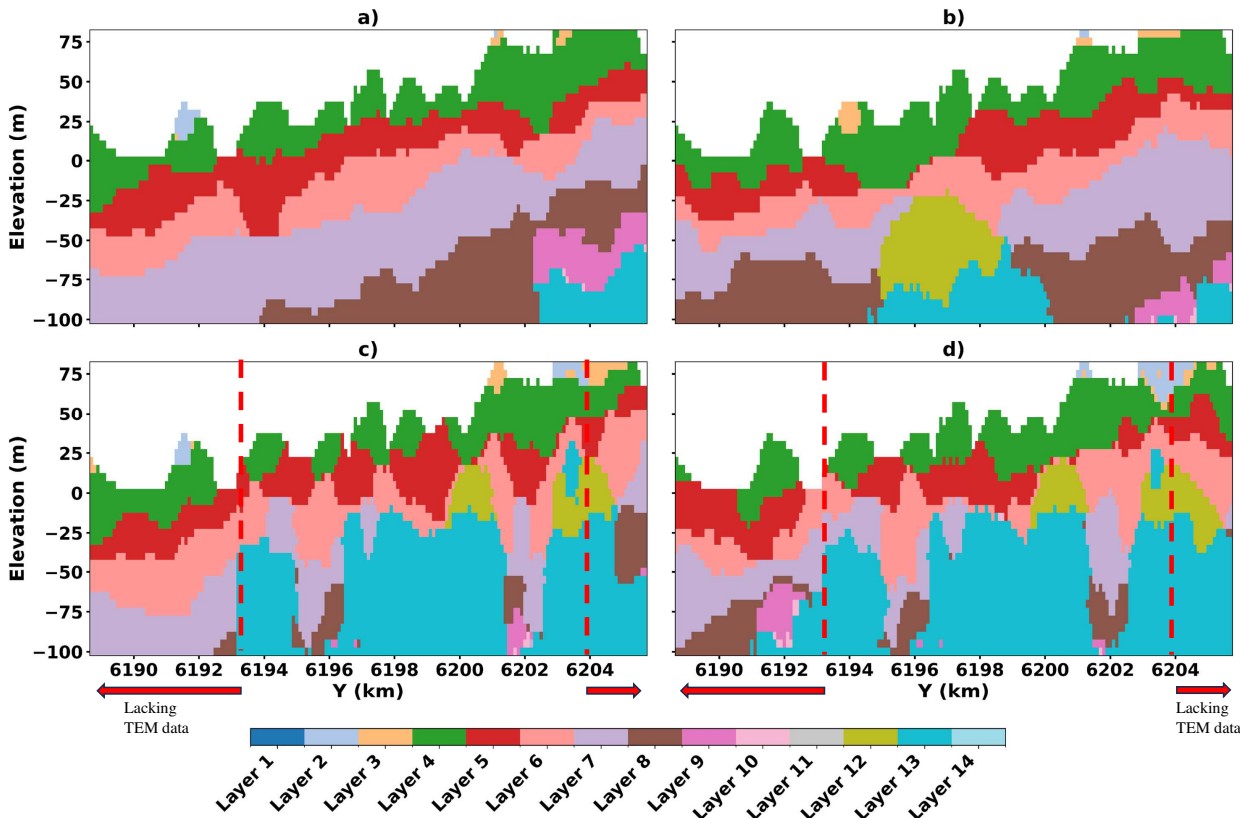

**Figure 12.** Modeling performance at transect of $X = 553600$ (UTM), red dash lines represents the boundary of existence of TEM data: **(a)** **(b)**: Two example realizations from MCP simulation without geophysical constraint. **(c) (d)**: Two example realizations from MCP simulation adding geophysical constraint.



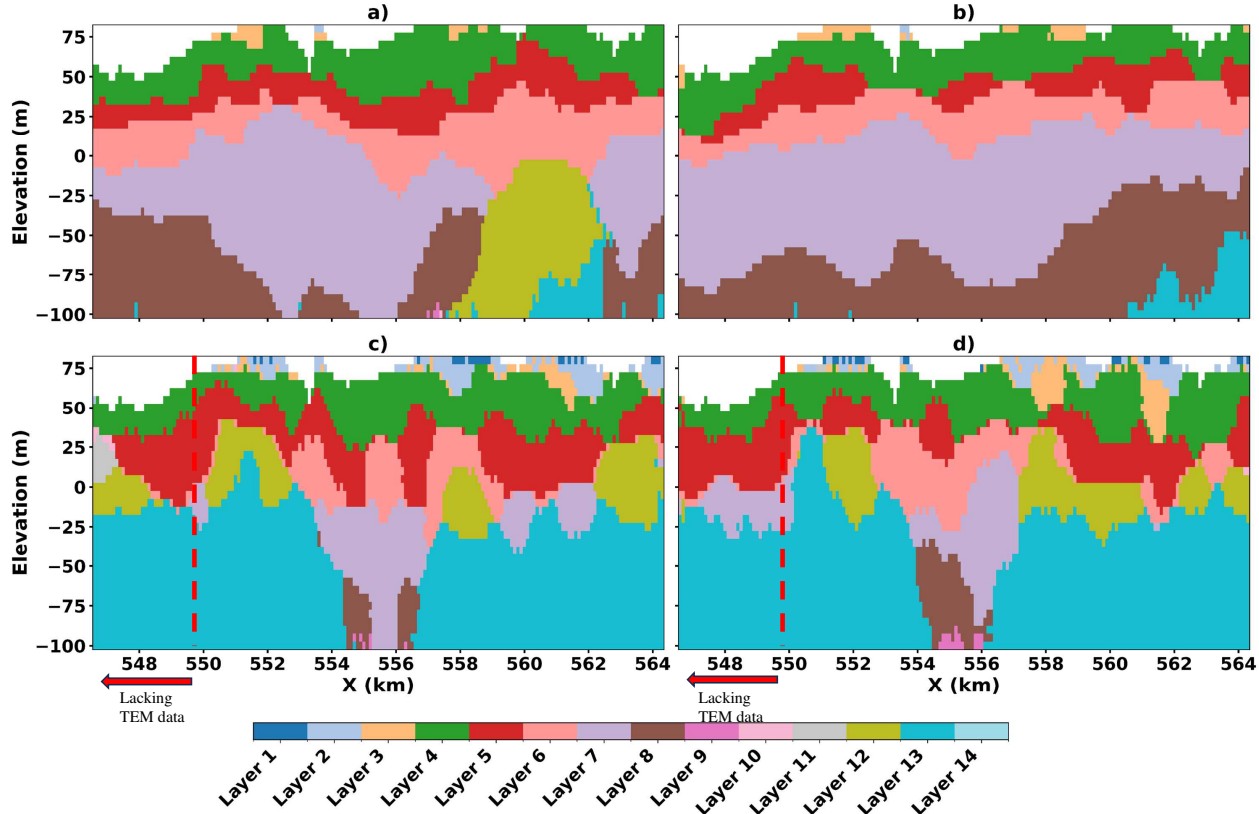

**Figure 13.** Modeling performance at transect of $Y = 6201300$ (UTM), red dash lines represents the boundary of existence of TEM data: **(a)** **(b)**: Two example realizations from MCP simulation without geophysical constraint. **(c)** **(d)**: Two example realizations from MCP simulation adding geophysical constraint.

To better evaluate the uncertainty of added geophysical constraints in our MCP simulations, we computed the expectation maps, entropy map which measure the diversity of predicted lithology categories at each location based on the Shannon entropy across realizations, and probability maps for the two transects, under both constrained and unconstrained scenarios.

The expectation of lithology at each grid location $(i, j)$ was computed as the average lithological value across all realizations:

$$\mathbb{E}[Z_{i,j}] = \frac{1}{N}\sum_{k=1}^{N} z_{i,j}^{(k)} \tag{10}$$

where $\mathbb{E}[Z_{i,j}]$ is the expected lithology at location $(i, j)$, $z_{i,j}^{(k)}$ is the lithological value at realization $k$, and $N$ is the total number of realizations.



The entropy at each grid location $(i,j)$ was calculated using Shannon's entropy to quantify the uncertainty in lithological prediction:

$$H(Z_{i,j}) = -\sum_{l=1}^{L} p_l(i,j) \log\left(p_l(i,j) + \varepsilon\right) \tag{11}$$

where $H(Z_{i,j})$ denotes the entropy at location $(i,j)$, $L$ is the total number of lithology classes, $p_l(i,j)$ is the empirical probability of class $l$ at location $(i,j)$ based on the realizations, and $\varepsilon$ is a small constant added to prevent numerical issues when $p_l(i,j) = 0$.

The expectation maps (Left of Figures 14 and 15) further confirm that incorporating constraint from inverted resistivity data sharpens the delineation of complex subsurface features and brings the realizations significantly closer to the existing geological model. The entropy maps (Right of Figures 14 and 15) show that, in the absence of sufficient hard data, the baseline MCP simulations exhibit widespread uncertainty across most of the domain. However, when geophysical constraints are incorporated, the uncertainty is reduced, especially in the deeper, thicker lithological units (e.g., layers 12 and 13). In particular, based on the probability maps of each layer at Figures 16 and Figures 17, the sand-rich zones in Layer 7 are more spatially continuous and correspond well with low-resistivity zones from the inverted AEM data. The probability distributions also reflect this improvement, with values for the dominant categories in each layer often exceeding 0.8 in well-constrained areas, indicating higher model confidence. Despite this improvement, the valley zones still exhibit higher entropy compared to more uniform lithological layers. This indicates the complexity of buried valley structures and highlights the need for higher resolution geophysical data to better resolve abrupt facies transitions. Importantly, the variability across realizations is representative of the uncertainty in the hydrostratigraphic structure of the reservoir, which can impact groundwater resources assessment.

It is important to note that the impact of geophysical constraints is not uniform across the transects as can be seen in the spatial distribution of boreholes and SkyTEM flight lines in the study area, see Figure 6. In areas where resistivity data coverage is dense, the MCP realizations show significant reductions in uncertainty and more geologically plausible transitions. However, in peripheral regions or at the edges of the model domain where resistivity coverage is sparse or absent (see indicated areas on Figure 14 and Figure 15), uncertainty remains high, and the model reverts to patterns driven primarily by the training image. What distinguishes the MCP framework, however, is its ability to transition smoothly between data-constrained zones and prior-based inference through its probabilistic treatment. This is made possible by its sequential simulation process, where simulated values in well-constrained areas are progressively used as new conditional data for simulating the remaining parts of the model. This mechanism allows local observations to gradually influence larger domains, providing a probabilistically coherent integration of observed data and prior geological structure.

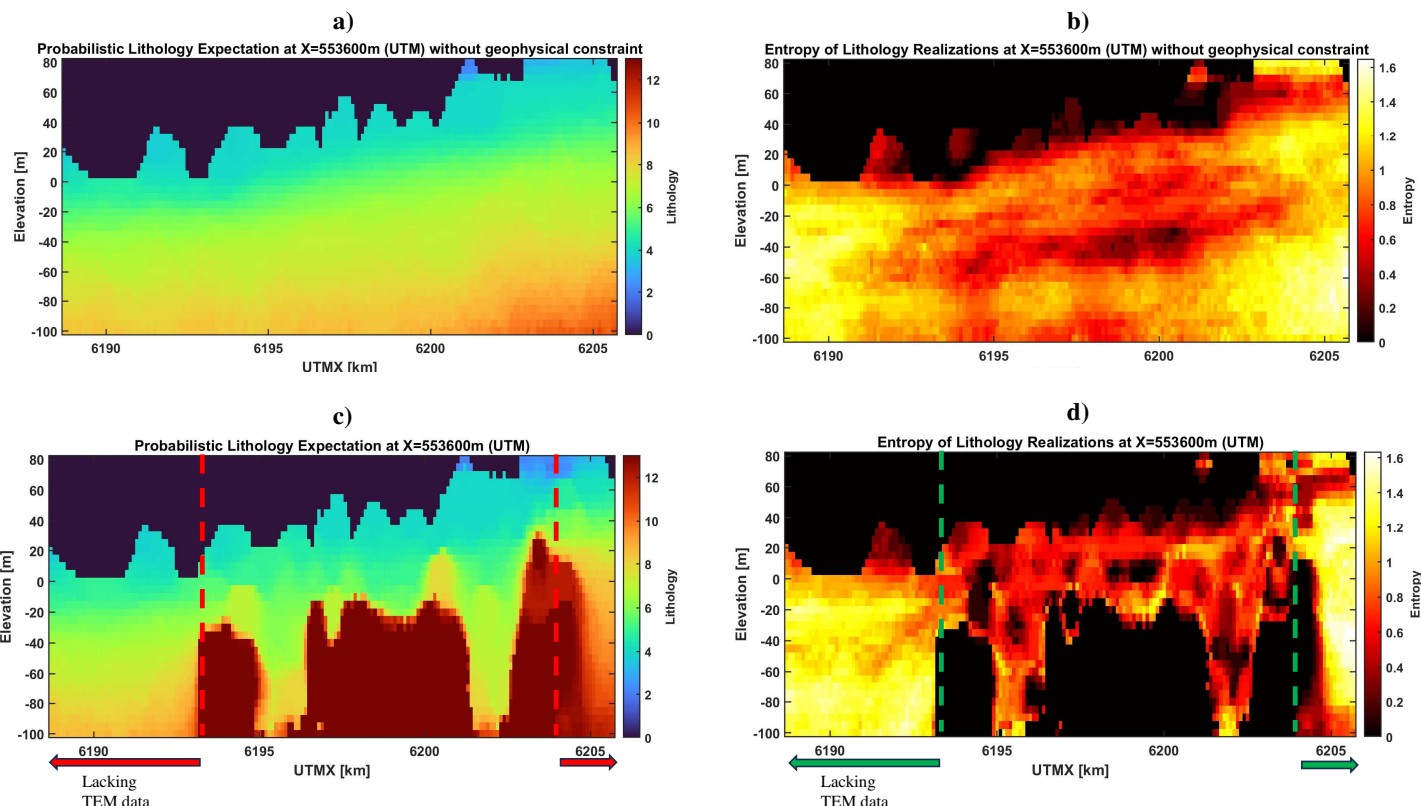

**Figure 14.** Uncertainty analysis at transect of $X = 553600$ (UTM), red dash lines represents the boundary of existence of TEM data: **(a)** **(b)**: Entropy and expectation maps based on MCP realizations without geophysical constraint. **(c)** **(d)**: Entropy and expectation maps based on MCP realizations adding geophysical constraint.





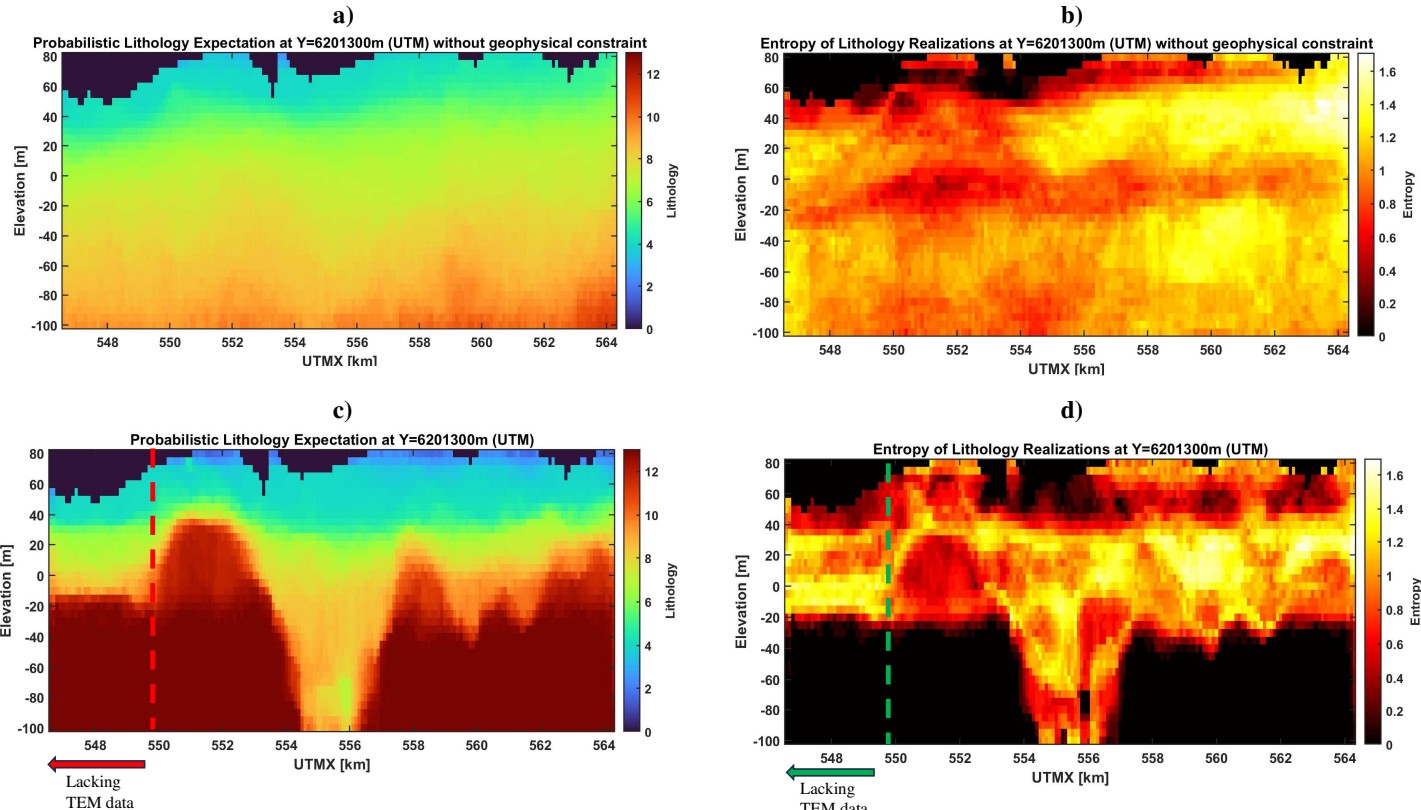

**Figure 15.** Uncertainty analysis at transect of $Y = 6201300$ (UTM), red dash lines represents the boundary of existence of TEM data: **(a)** **(b)**: Entropy and expectation maps based on MCP realizations without geophysical constraint. **(c)** **(d)**: Entropy and expectation maps based on MCP realizations adding geophysical constraint.



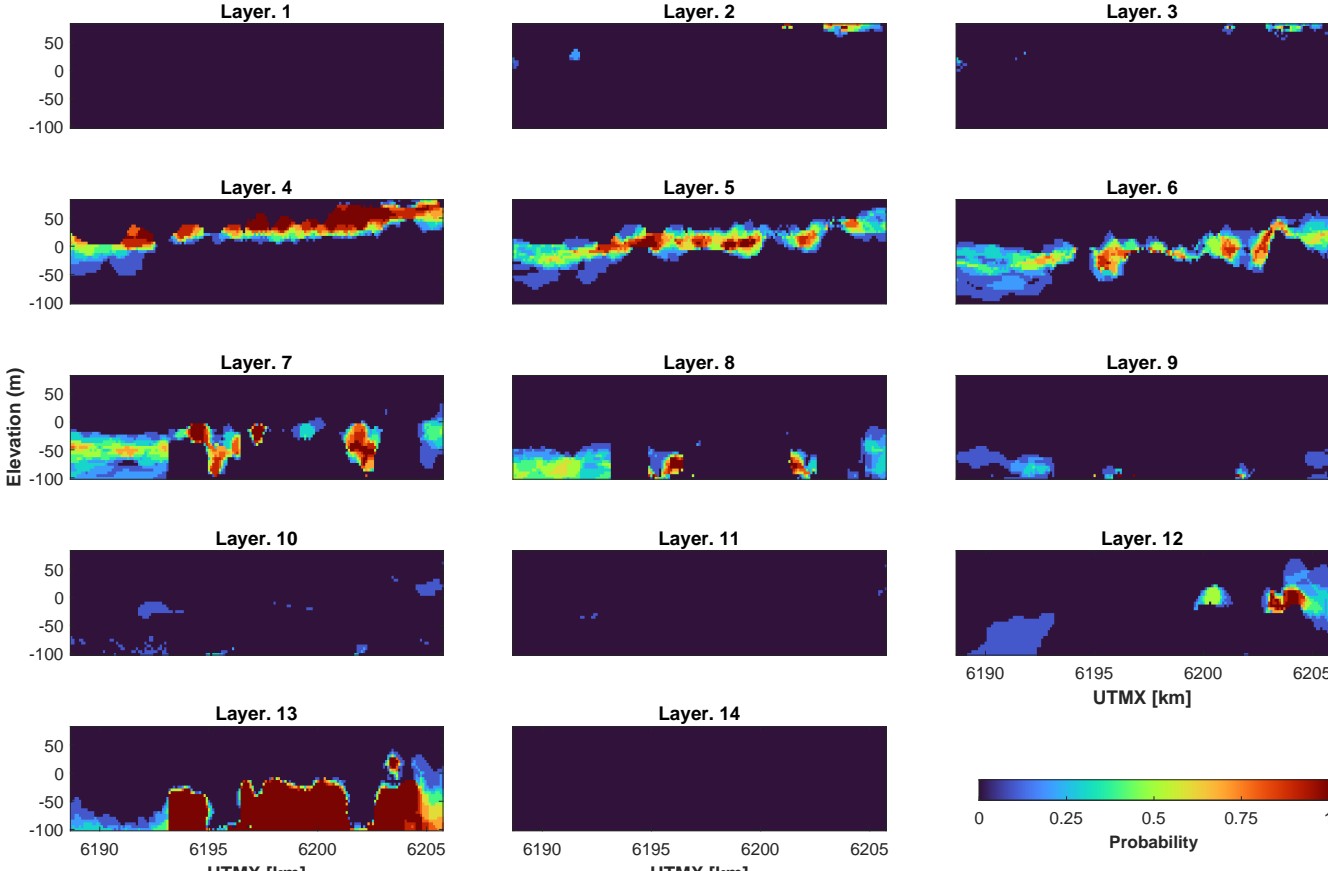

**Figure 16.** Probability distribution of each lithological layer in each pixel from 100 MCP simulations with geophysical constraint at transect of X = 553600 (UTM)





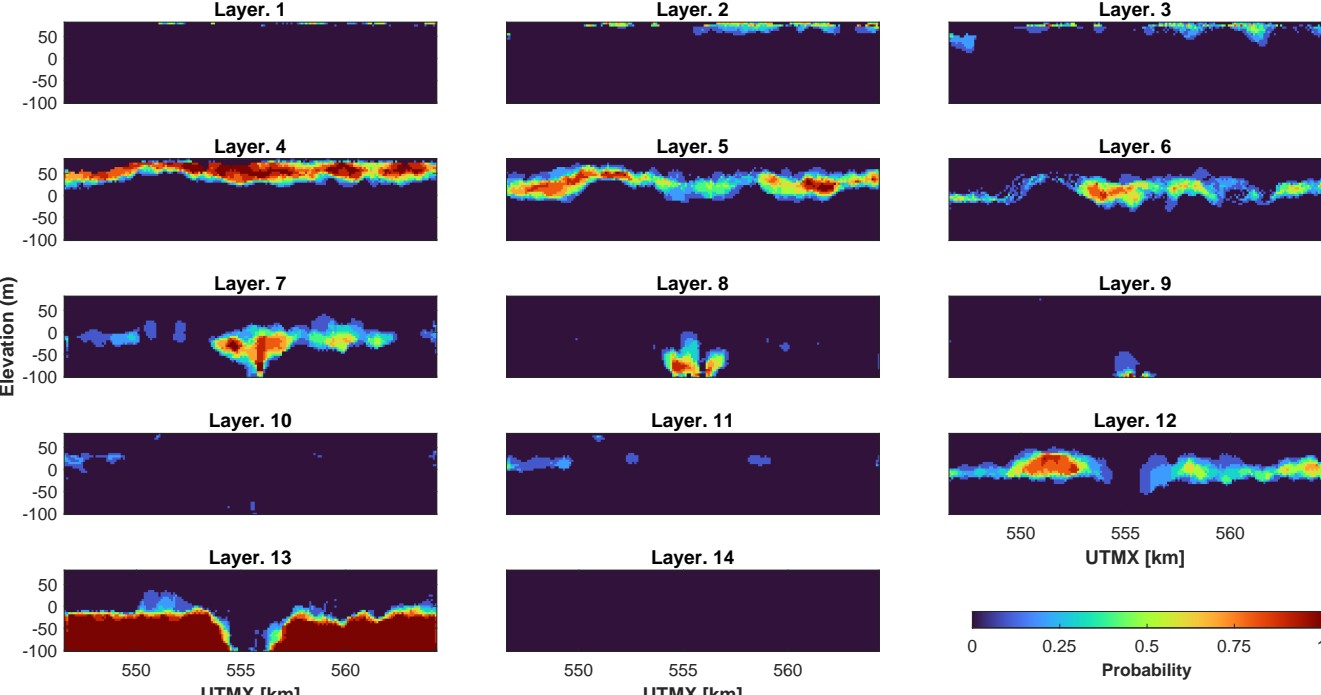

**Figure 17.** Probability distribution of each lithological layer in each pixel from 100 MCP simulations with geophysical constraint at transect of Y = 6201300 (UTM)

## 4 Discussion

To better understand how our novel MCP framework with geophysical constraint better captures lithological proportions under geophysical constraints, a comparison plot was produced using the transect at $Y = 6201300$ as an example (Figure 18). It evaluates the consistency and variability in the simulated lithology proportions across multiple realizations, and how these compare with the training images and the lithological composition of the target transect. The overall facies proportions derived from the training images are generally comparable to those realizations obtained with geophysical constraints, suggesting broad

consistency in global distributions. However, several lithological units show noticeable inconsistencies between the simulated proportions and those derived from the training images. For example, Layer 4 and Layer 12 are clearly overrepresented in the MCP realizations compared to their occurrence in the training images. This overestimation is likely influenced by a few deep boreholes in the transect area, causing a strong local constraint during simulation. In contrast, Layers 6, 7 and 8 appear to be underrepresented in the simulations. These lithologies are associated with deep valley structures in the geological model.

However, due to the limited number of deep boreholes in these areas, the available hard conditioning data are sparse or entirely absent. In the absence of geophysical data, the probability values derived from framework are insufficient to capture and preserve the complex geometry of these buried valleys. The framework tends to favor a more uniform stratification, leading to a reduced occurrence of these deeper facies in the realizations (Benoit et al., 2018). Nevertheless, the integration of geophysical





constraints helps to simulate these structures. While it does not significantly alter the proportions, it successfully limits unreal-

istic deviations and maintains reasonable spatial consistency in the valley zones.

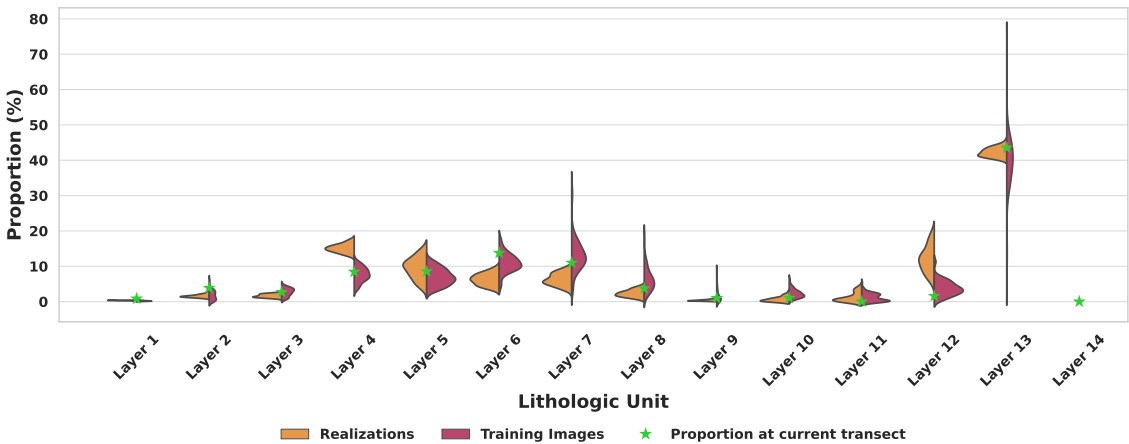

**Figure 18.** The yellow bars left side show the distribution of proportions across multiple MCP simulation realizations with constraint from geophysical data, the red bars right side represent the distribution derived from the 88 different training images, and the star markers indicate the mean proportion of each lithology observed in the current target transect from interpreted model.

It is essential to examine closer the role of the choice of TI and how it influences the outcomes. A sensitivity analysis aims to evaluate the robustness of the MCP framework with respect to different TI selections and to determine the extent to which lithological predictions depend on the chosen TI.


To perform such analysis, 11 transects were randomly selected from the 88 available transects to serve as individual train-ing images. For each selected TI, 100 realizations of the lithological model were generated using the MCP framework with geophysical constraints. The Jaccard dissimilarity index (Equation (7)) was calculated for each realization with respect to the interpreted geological model, providing a quantitative measure of dissimilarity between the predicted and reference models.

In addition to these random selections, an integrated TI, constructed by combining information from all 88 transects, was used to produce another set of 100 realizations. Furthermore, a separate scenario was tested in which the target transect itself ($Y = 6201300$ UTM) from the interpreted model was used as the training image. This setting, referred to as the 'Best TI' in the results, uses as an idealized case where the training data perfectly reflect the spatial patterns of the target.

The results of this sensitivity analysis are presented in the Figure 19, which displays the average Jaccard dissimilarity index for all 13 TI (Training Image) scenarios (11 random TIs, 1 integrated TI and one the target transect itself). Across all scenarios, the average dissimilarity values are remarkably similar, with minimal variation observed. This indicates that the lithological predictions remain consistent regardless of the TI selection. This observation is consistent with findings by Barfod et al. (2018),




who used multiple-point statistics in combination with SkyTEM data and noted that the influence of the training image was significantly reduced when strong geophysical constraints were applied. In contrast, under the low geophysical constraint scenario, the influence of TI selection becomes larger. However, the overall variation in results remains small. This is primarily because the boreholes in the study area are shallow and provide minimal hard conditioning data. Consequently, the simulations generated using only the MCP framework, without incorporating geophysical information, rely largely on bivariate transition probabilities. These probabilities inherently preserve the expected stratigraphic ordering (e.g., older layers occurring at greater depths), leading to realizations that exhibit relatively uniform layering and limited structural variability. Notably, similar results were obtained when using an integrated TI compared to using the target transect alone, indicating that the ELOP based integration strategy performs reliably within the MCP framework.

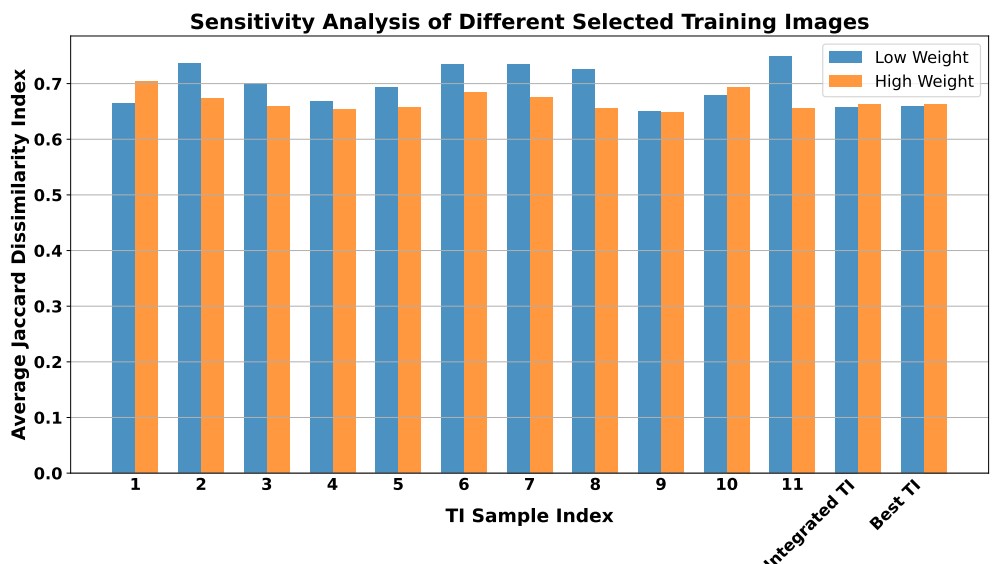

**Figure 19.** Sensitivity analysis of Training Image (TI) selection on lithological predictions using the MCP framework with geophysical constraints. The bar plot presents the average Jaccard dissimilarity index computed over 100 realizations for each of the 13 TI scenarios. These include 11 randomly selected TIs, one "Integrated TI" constructed from all 8 transects, and one "Best TI" using the current transect as the TI. Blue bars represent results under a low weight of geophysical constraint, while orange bars represent results under a high weight of geophysical constraint.

An important factor that influences the quality of lithology simulations is the way geophysical resistivity data are acquired and processed (Rochlitz et al., 2023; Javed et al., 2019). In this study, the resistivity model was generated through the Laterally Constrained Inversion (LCI) method from airborne electromagnetic (EM) data and then interpolated along borehole paths to estimate continuous resistivity values with depth. Both steps introduce a degree of smoothing: the inversion averages resistivity over vertical windows, which tends to reduce sharp boundaries between lithological units, while interpolation smooths



spatial variations further due to the distance between flight lines and boreholes. As a result, the resistivity values used to link

with lithological categories do not fully capture the fine-scale variations of the true subsurface and may present a simplified or blurred version of reality. This smoothing directly affects the conditional probability distributions used in the MCP simulations. When resistivity contrasts between lithologies become less distinct, especially at depths where their values overlap, the probability curves become wider and their peaks less defined. This increases the uncertainty in lithology classification during simulation, particularly in zones where several units share similar resistivity ranges. More advanced inversion techniques

(Deleersnyder et al., 2023) could potentially improve conditional probability from geophysical data as suggested in Hermans and Irving (2017).

In parts of the model where deep boreholes data are missing, the simulation process becomes more uncertain. Without hard information to constrain the deeper layers, MCP relies mainly on the training image and the overall proportion statistics, which

may not reflect the specific conditions of the target area (Benoit et al., 2018). In this situation, the use of geophysical data becomes particularly helpful. Even though the resistivity model is smoothed due to inversion and interpolation, it still carries important signals about the general layering and structural trends at depth. By incorporating this resistivity information, the conditional probabilities in MCP are guided by more than just borehole data and facies proportions. This helps the simulation pick up on major features, such as buried valleys or sharp transitions that would likely be missed or misrepresented otherwise.

As a result, geophysical constraints play a key role in maintaining geological plausibility in deeper zones where direct observations are lacking.

While no direct MPS simulations were performed in this study, a relevant comparison can be drawn from the work of Barfod et al. (2018), who applied MPS simulations in the same Egebjerg field site. In their approach, airborne electromagnetic

(SkyTEM) data were first inverted and then transformed into categorical probability maps, which served as soft conditioning data in a direct sampling MPS framework. This allowed the geophysical constraints to influence the final model, but only indirectly, by treating the inverted resistivity as a secondary training image, rather than incorporating it through explicit probability distributions. By contrast, the MCP framework used in this study relies on statistically derived bivariate transition probabilities extracted from interpreted geological models. It also enables the combination of directional transition probabilities from mul-

tiple transects using the Extended Logistic Opinion Pool (ELOP) approach, allowing for a consistent probabilistic framework across directions. More importantly, MCP provides a transparent mechanism for integrating soft geophysical constraints via conditional probability functions, which capture the uncertainty, smoothing effects, and resolution loss associated with geophysical inversion. Unlike Direct Sampling MPS, which incorporates soft data through auxiliary images, the MCP framework treats soft constraints probabilistically and explicitly, making it particularly well suited for vertically ordered, non-stationary

hydrostratigraphic settings. This flexibility in combining hard borehole data with soft geophysical information makes one of the key advantages of MCP in such contexts. For the Egebjerg field site, Madsen et al. (2022) also addressed uncertainty, but within the framework of a deterministic conceptual model primarily guided by expert interpretation and constrained geological assumptions. In contrast, the MCP based simulations presented in this study offers a probabilistic representation of uncer-





tainty, grounded in data driven inference and stochastic modeling. Unlike sequential layer-based modeling which may require layer-by-layer corrections, MCP accounts for vertical dependencies through transition probabilities. This allows the model to naturally preserve stratigraphic logic during simulation, although at the cost of requiring post-processing after incorporating geophysical constraints, which slightly alters the posterior distribution. Nevertheless, the resulting realizations are more informative for probabilistic groundwater modeling.

Future work could benefit from incorporating more advanced inversion techniques. While the current approach relies on deterministic 1D and laterally constrained inversion methods, adopting new inversion strategies could reduce smoothing artifacts and preserve sharper resistivity contrasts (Deleersnyder et al., 2023), thereby improving the quality of the resistivity–lithology relationships used in simulation. Furthermore, the observed robustness to TI selection opens the possibility of using geological models, such as the interpreted 3D model of the Egebjerg catchment, as generalized training images for simulating similar geological settings in neighboring regions. This would allow for more efficient modeling efforts over a wider spatial scale without the need to generate customized training images for each new site. However, this generalization relies heavily on the availability of dense and high-quality geophysical data (e.g., SkyTEM). In addition, the post-processing corrections contribute to how informative the geophysical constraint becomes. These choices play a crucial role in enabling the geophysical data to compensate for potential mismatches between the TI and the target area.

## 5 Conclusion and Prospective

Hydrostratigraphic modelling is a complex task as it requires to interpret information from sparse boreholes, leading to large uncertainty. In this context, geophysical data brings spatially distributed but indirect information to help interpolating stratigraphic units across boreholes. In this study, we developed and demonstrated an enhanced Markov-type Categorical Prediction (MCP) framework that integrates airborne electromagnetic (AEM) geophysical data as soft probabilistic constraints for lithological realization generation. Through both synthetic and real-world case studies, we demonstrated that the inclusion of geophysical information significantly enhances the geological realism of simulated subsurface models, while largely preserving their spatial continuity, particularly in areas with sparse borehole coverage. The MCP framework retains its strength in enforcing stratigraphic ordering, while the geophysical integration helps guide the simulation toward more plausible large-scale structures such as buried valleys. Geophysical data are integrated under the form of conditional probability obtained from comparing AEM inverted resistivity with borehole data. To improve the efficiency of the framework, we derive depth-dependent conditional probability, allowing the framework to maximize the information content brought by geophysics and seaminglessly integrate it in MCP. As an output, the framework provides a series of geological realizations consistent with the expected hydrostratigraphic layering and the geophysical data from which conditional probabilities of lithologies can be derived. The probabilities vary between well resolved zones close to boreholes where the probability of lithology is high, to less-well resolved zone where the geophysical data leaves a larger uncertainty and several lithologies have similar probabil-

ities. This variability across realizations can be used to target new sampling locations or as input to dynamic simulations for groundwater resource management.

In summary, the enhanced MCP framework presented in this study offers a flexible, efficient, and geologically informed approach for subsurface modeling in data-limited environments. By unifying hydrostratigraphic constraints, borehole logs, and geophysical observations in a single probabilistic framework, the method bridges the gap between MCP geostatistics and integrated geophysical data modelling, and provides a strong foundation for future developments in uncertainty aware geological simulation.

*Code availability.* The simulation codes used in this study are proprietary and were made available to the authors through collaboration. Access to the code can be granted upon reasonable request by contacting the corresponding author.

*Data availability.* The constructed 3D hydrostratigraphic model used in real-case study is openly available in the GEUS Dataverse at (https://dataverse.geus.dk/dataset.xhtml?persistentId=doi:10.22008/FK2/FHP1XK). The resistivity data is accessible through the open-access national geophysical database GERDA (https://eng.geus.dk/products-services-facilities/data-and-maps/national-geophysical-database-gerda),
and borehole locations and descriptions are available in the open-access national borehole database JUPITER (https://eng.geus.dk/products-services-facilities/data-and-maps/national-well-database-jupiter).

*Author contributions.* LM implemented the workflow, developed the simulation code, carried out the data processing, and prepared the initial draft of the manuscript. WD, TH, DD, and EV provided continuous supervision, with critical input on the methodological design and interpretation of the results. NB supported the original code and contributed to the code revision. RM and JN supplied the field data and
assisted with the processing and critical assessment of the results. All authors contributed to the discussion of the results and to the final version of the manuscript.

*Competing interests.* The authors declare that they have no competing interests.

*Acknowledgements.* We gratefully acknowledge the Geological Survey of Canada and other partners for granting access to their proprietary code, with the support of Nicolas Benoit. This research was supported by funding from the KU Leuven Postdoctoral Mandate
(PDMT2/23/065). The first author also acknowledges financial support from the Ghent University Research Fund (BOF.BAF.2024.0073.01) and the China Scholarship Council for doctoral studies.



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
