# Peer review of "Enhanced Markov-Type Categorical Prediction with Geophysical Soft Constraints for Hydrostratigraphic Modeling"

_EGUsphere, 2025_

## Author Comment (AC1)

We sincerely thank this reviewer for the valuable comments. Our detailed responses and corresponding revisions are provided below. The final revised manuscript will incorporate these and the comments from other reviewers.

**Introduction:**

**1.**
**Comment: Line 59, the reference to Mariethoz et al. 2010 (Direct Samling specific algorithm, as mentioned line 551) might not be the most appropriate to support your statement here. Guardiano and Srivastava 1993 instead?**

Reply to the reviewer: We recognize that MPS was first proposed by Guardiano and Srivastava (1993) and the reference was added. However, we also think it is important to propose a more recent reference for MPS, to guide the readers towards successful applications.

**Original:** '*Multiple-Point Statistics (MPS) has become a widely used geostatistical method in hydrostratigraphical modelling. MPS uses Training Images (TIs) to quantify the spatial variability and reproduce complex geological patterns that cannot be captured by traditional two-point geostatistics (Mariethoz et al. 2010)*'

**Revision:**'*Multiple-Point Statistics (MPS) has become a widely used geostatistical method in hydrostratigraphical modelling. MPS uses Training Images (TIs) to quantify the spatial variability and reproduce complex geological patterns that cannot be captured by traditional two-point geostatistics (Guardiano and Srivastava, 1993; Mariethoz and Caers, 2014)*'

**2.**
**Comment: Line 69: what do you mean by that? How does the reference to Meerschman et al. 2013 support this statement? This reference would better describe the ease of use or wide use of MPS techniques such as the Direct Sampling.**

Reply to the reviewer: It is right that Meerschman et al. (2013) did not treat specifically the case of geophysics. The sentence was revised, and a reference to Lochbühler et al. (2014) was added to support the statement.

**Original:** '*Moreover, the integration of geophysical data is often heuristic and lacks a formally probabilistic structure (Meerschman et al., 2013).*'

**Revision:**'*Most existing MPS approaches, such as Direct Sampling (Meerschman et al., 2013), still treat soft or geophysical data in a heuristic manner, through secondary training images rather than probabilistic conditioning (e.g., Lochbühler et al., 2014), so they cannot explicitly address vertical non-stationarity within a probabilistic framework.*'

**3.**

**Comment: Many statements refer to 4 different citations. Maybe keep the two most relevants and cite as (e.g. …).**

Reply to the reviewer: Thanks for noticing this. We modified the text accordingly.

**Original:** '*Incorporating uncertainty into the simulation of geological heterogeneity, geostatistical approaches provide not only plausible geological scenarios but also essential input for ensemble-based hydrogeological forecasting, which is one type of probabilistic approach that relies on multiple realizations to assess model uncertainty. (Moore et al., 2022; Zimmerman et al., 1998; Enemark et al., 2024; Hermans et al., 2015).*'

**Revision:** '*Incorporating uncertainty into the simulation of geological heterogeneity, geostatistical approaches provide not only plausible geological scenarios but also essential input for ensemble-based hydrogeological forecasting, which is one type of probabilistic approach that relies on multiple realizations to assess model uncertainty. (e.g. Enemark et al., 2024; Hermans et al., 2015).*'

**4.**

**Comment: Relevant work that could probably be included in the literature review:**
**Lochbühler, T., Pirot, G., Straubhaar, J., & Linde, N. (2014). Conditioning of multiple-point statistics facies simulations to tomographic images. Mathematical Geosciences, 46(5), 625-645.**
**Pirot, G., Linde, N., Mariethoz, G., & Bradford, J. H. (2017). Probabilistic inversion with graph cuts: Application to the Boise Hydrogeophysical Research Site. Water Resources Research, 53(2), 1231-1250.**

**Original :** '*Madsen et al. (2021) proposed treating uncertain geological interpretations as probabilistic constraints, comparing MPS and Gaussian simulation methods, and showed that MPS produced more geologically plausible and connected realizations. Hermans et al. (2015) developed a full MPS-based inversion framework that used ERT data both to falsify prior geological scenarios and to locally constrain groundwater simulations, showing the strength of MPS in quantifying uncertainty and integrating multiple data types.*'

'*Although geophysical data do not directly measure lithology, they provide property contrasts (e.g., in resistivity) that, after inversion and interpretation, can be statistically linked to hydrofacies distributions (Michel et al., 2020; Looms et al., 2008).*'

**Revision:** '*Madsen et al. (2021) proposed treating uncertain geological interpretations as probabilistic constraints, comparing MPS and Gaussian simulation methods, and showed that MPS produced more geologically plausible and connected realizations. Hermans et al. (2015) developed a full MPS-based inversion framework that used ERT data both to falsify prior geological scenarios*

*and to locally constrain groundwater simulations, showing the strength of MPS in quantifying uncertainty and integrating multiple data types. Lochbühler et al. (2014) demonstrated that tomographic images can be used to condition multiple-point statistics facies simulations, thereby improving the structural consistency of simulated geological models.'*

**'***Although geophysical data do not directly measure lithology, they provide property contrasts (e.g., in resistivity) that, after inversion and interpretation, can be statistically linked to hydrofacies distributions (Michel et al., 2020; Pirot et al. 2017).***'**

**5.**

**Comment: You have to clarify in the introduction that you are building up on previous work (Isunza Manrique et al., 2023) or ideas presented at a conference (Guo et al. 2024) and explain what is new here.**

Reply to the reviewer: A preliminary version of this study was presented by us at a conference (Guo et al., 2024), where the integration of geophysical inversion with the MCP framework was demonstrated using a synthetic case. The present manuscript extends this framework to a real-world 3D geological setting and includes detailed quantitative assessments of uncertainty reduction and model sensitivity. Although the principle of permanence of ratios has also been applied by Isunza Manrique et al. (2023) to combine probabilistic information from multiple geophysical attributes, specifically resistivity and chargeability, their approach focuses on probabilistic interpretation of inversion results and does not include geostatistical simulations. In contrast, the present study extends the work of Benoit et al. who developed the MCP framework and use the permanence of ratio to formally integrate probabilistic geophysical information into the MCP realizations.

**Original:** '*A recently applied geostatistical approach by Benoit et al. (2018), known as Markov-type Categorical Prediction (MCP), provides an alternative framework to traditional multiple-point statistics (MPS) for simulating categorical geological units. MCP uses bivariate transition probabilities derived from a training image. One of the key advantages of MCP is that it reduces the dependence on high-quality or highly repetitive training images, which can be a limiting factor in some MPS implementations (Allard et al., 2011).. When key features in the TI are sparse, irregular, or unique, MPS may struggle to reproduce them consistently, potentially leading to artificial discontinuities or oversimplified realizations (Barfod et al., 2018). By contrast, MCP operates on a different principle. Rather than trying to reproduce entire patterns from the TI, MCP uses pairwise transition probabilities between units to capture the likelihood of one unit being adjacent to another (Benoit et al., 2018). This approach allows MCP to extract essential geological information in a non-stationary fashion without needing a complete TI. Furthermore, MCP remains computationally efficient, even when simulating models with a large number of lithological categories because it avoids high-order pattern scanning or search-tree construction. One of MCP's strengths is its ability to strictly respect geological rules when certain transitions between units are geologically impossible. For example, if a specific lithological unit is never observed directly above another in the training data, MCP ensures that this configuration will not appear in the simulated model based*

*on zero bivariate probability of these two units (Benoit et al., 2018). Yet, previous applications of the MCP framework have relied almost exclusively on hard conditioning data, such as borehole lithology. In settings where such data are sparse, the method often defaults to random simulation, which can result in geologically unrealistic outputs (Benoit et al., 2018). However, MCP offers greater transparency and flexibility in conditioning, making it well suited for the integration of soft information derived from geophysical inversion models. To leverage this potential, our study extends the MCP framework by incorporating geophysical soft constraints into the simulation process. This integration aims to reduce uncertainty and enhance the geological realism of subsurface models, particularly in areas that are poorly constrained by hard data.'*

**Revision:** Insert the following paragraph after the one above: '*A preliminary version of this work was presented at a conference (Guo et al., 2024), where the feasibility of integrating geo-physical inversion results into the MCP framework was demonstrated using a synthetic case. Building on that foundation, the present study extends the approach to a real-world 3D geological setting and includes a comprehensive quantitative evaluation of uncertainty reduction and model sensitivity. In this work, soft information derived from geophysical inversion models is merged into the MCP framework using the principle of permanence of ratios from the study of Isunza Manrique et al. (2023), where they combined probabilistic information from multiple geophysical attributes such as resistivity and chargeability. However, their work was limited to the probabilistic interpretation of inversion results without performing geostatistical simulations. In contrast, our study integrates this principle directly within the MCP simulation process, enabling the formal incorporation of probabilistic geophysical information into the categorical realization generation. This represents a methodological advancement over existing MCP applications by linking geostatistical simulation with geophysically derived probability fields, thereby improving both geological realism and interpretability of subsurface models.'*

**Method:**

**6.**
**Comment: 2.1 MCP: Is the considered lag h omnidirectional or directional?**

Reply to the reviewer: In the MCP algorithm, the lag $h=(hx, hy, hz)$ represents directional lag vector between the simulated node and its neighboring nodes. For each pair of facies categories (i,j), the precomputed matrix $gh_{ij}(hx, hy, hz)$ stores the bivariate probabilities as a function of this lag. During simulation, for each node, the algorithm retrieves the corresponding probability values from $gh_{ij}$ according to the relative vector (dx,dy,dz) to neighboring data and combines them (equation 1) to update the conditional probability of the current facies.

**Original:**

**Revision:**

**7.**

**Comment: Line 167: 'this' is ambiguous. Do you refer to Guo et al. 2024 or the work presented here?**

Reply to the reviewer: Thanks for pointing this out. We clarified the sentence.

**Original:** '*Geophysical data can provide additional constraints to geostatistical simulations by linking lithological categories with physical properties (Guo et al., 2024).* **In this study,** *a stochastic resistivity–lithology relationship is established by deriving conditional probabilities from inverted resistivity models.*'

**Revision:** '*Geophysical data can provide additional constraints to geostatistical simulations by linking lithological categories with physical properties (Guo et al., 2024).* **In the present study**, *a stochastic resistivity–lithology relationship is established by deriving conditional probabilities from inverted resistivity models.*'

**8.**

**Comment: It is not clear how P(A|C) is estimated nor how P(A|B,C) is integrated in the MCP framework (equation 1).**

Reply to the reviewer: $P(A|C)$ was derived from the inverted resistivity models through lithology–resistivity calibration. For the synthetic case, the entire training image (TI) was used to establish this relationship, whereas for the real-field case, borehole data were employed for calibration. During simulation, for a unsimulated node, the MCP algorithm first computes the conditional probability $P(A|B)$, as according to equation 1, based on the bivariate probabilities derived from neighboring points. This probability is then immediately merged with the lithology–resistivity conditional probability $P(A|C)$ using the permanence-of-ratios formulation (equation 5) to obtain $P(A|B,C)$, which is subsequently used for random sampling to determine the lithofacies at that node.

**Original:**

**Revision:** The text was modified accordingly below equation 5.

**9.**

**Comment: Line 231: what are the different variables composing the training image? (maybe insert a step between 4. And 5.).**

Reply to the reviewer: We are not entirely certain that we understood this comment as intended.

Since the calculation of the joint probability has been clarified in the previous comment, we assume the question is related on how the training image was selected. For training image of the synthetic case described in Line 23, a manually synthetic three-layer lithological model was constructed on a 2D 80 × 50 grid. The training image (TI) used in the synthetic case corresponds to this simplified three-layer model, containing three categorical facies.

**Original:**

**Revision:**

**10.**

**Comment: Figure 1 is confusing; there are two steps 6, crossing arrows, please reorganise it to make it clear or remove if the text description above is clear enough. Then later comes Figure 11, that looks totally different. It would be better to have a single workflow figure in section 2, and then give the specific of how the TI and conditional probabilities are estimated for the synthetic case and the real-case study.**

Reply to the reviewer: A general workflow illustrating the constrained MCP working principle and the integration of geophysical data has been added at the end of Section 2. This figure provides an overview of how the MCP algorithm operates and how soft geophysical constraints are incorporated. Since the synthetic case and the real-field case have different objectives and slightly different workflow, we decided to keep a figure for both. For the synthetic case, the approach involves using the true lithological model to generate the synthetic conductivity distribution, applying a forward model to generate TEM sounding data, and performing a 1D inversion to obtain the conductivity model. For the real-field case, geophysical data already exist in this area, and we use lithological information to statistically link the lithology and resistivity data. We actually implemented this in a '3D' manner: multiple profiles were used as training images to derive bivariate probabilities, which were then merged into a general one containing more information, and finally applied to simulate multiple locations.

**Original:**

**Revision:** Provide a general workflow for the constrained MCP working principle at the end of Section 2.:

[Figure]

Also, the workflow plots of both the synthetic and field cases have been improved for clarity according to the comment.

Synthetic case:

[Figure]

Real-field case:

[Figure]

**Results and discussion:**

**11.**
**Comment: Figure 9a: use the same colormap as in Figure 7.**
**Figure 9b: use a perceptually uniform colormap (e.g. https://doi.org/10.1038/s41467-020-19160-7 , https://www.fabiocrameri.ch/colourmaps/ , https://colorcet.com/ )**

Reply to the reviewer: Thanks very much for the helpful suggestion to improve the readability of the figures. The figures have been revised accordingly.

**Original:**

[Figure]

**Figure 9. (a):** Lithological information of each borehole. **(b):** Inverted resistivity values are interpolated along the borehole locations. **(c):** Inverted resistivity distribution with each lithology type based on interpolated resistivity values.

**Revision:**

[Figure]

**Figure 9. (a):** Lithological information of each borehole. **(b):** Inverted resistivity values are interpolated along the borehole locations. **(c):** Inverted resistivity distribution with each lithology type based on interpolated resistivity values.

**12.**

**Comment: Line 440-441, maybe add a reference to support the use of Shannon's entropy, e.g. one of the followings:**

**1. Lindsay, M. D., Aillères, L., Jessell, M. W., de Kemp, E. A., & Betts, P. G. (2012). Locating and quantifying geological uncertainty in three-dimensional models: Analysis of the Gippsland Basin, southeastern Australia. Tectonophysics, 546, 10-27.**

**2. Pirot, G., Joshi, R., Giraud, J., Lindsay, M. D., & Jessell, M. W. (2022). loopUI-0.1: indicators to support needs and practices in 3D geological modelling uncertainty quantification. Geoscientific Model Development, 15(12), 4689-4708.**

**Line 445: averaging lithological categorical values seems dangerous. It may convey false information. E.g. if lithologies 10 (aquifer) and 12 (aquifer) average to 11 (aquitard), that would not make sense. It would make more sense to have an aquitard probability volume and an aquifer probability volume.**

Reply to the reviewer: The suggested reference is added to support the use of Shannon's entropy. We agree with the reasoning that if the interpretation is made in terms of properties of the layers, this could be misleading. However, in this case, we limit ourselves to the hydrostratigraphy itself. Since the layers are chronologically ordered, the average value is still meaningful in that sense. Nevertheless, both the expectation and entropy results demonstrate the same thing: the presence of the valley with geophysical constraints and reduced uncertainties. Therefore, it is sufficient to retain only the entropy results, and the average was removed.

**Original:**

For the uncertainty analysis of the real-field case, three types of plots were generated: expectation plots, entropy plots, and probability plots. However, the reviewer noted that producing expectation plots by averaging categorical classes can be misleading.

[Figure]

**Figure 15.** Uncertainty analysis at transect of $Y = 6201300$ (UTM), red dash lines represents the boundary of existence of TEM data: **(a)** **(b)**: Entropy and expectation maps based on MCP realizations without geophysical constraint. **(c)** **(d)**: Entropy and expectation maps based on MCP realizations adding geophysical constraint.

**Figure 14.** Uncertainty analysis at transect of $X = 553600$ (UTM), red dash lines represents the boundary of existence of TEM data: **(a)** **(b)**: Entropy and expectation maps based on MCP realizations without geophysical constraint. **(c)** **(d)**: Entropy and expectation maps based on MCP realizations adding geophysical constraint.

**Revision:** '*To better evaluate the uncertainty of added geophysical constraints in our MCP simulations, we computed entropy maps which measure the diversity of predicted lithology categories at each location based on the Shannon entropy (Pirot et al., 2022) across realizations, and probability maps for the two transects, under both constrained and unconstrained scenarios.*'

[Figure]

**Figure 14.** Uncertainty analysis based on entropy values, green dash lines represents the boundary of existence of TEM data: **(a)** **(c)**: Entropy maps based on MCP realizations along the transect at $X = 553600$ (UTM). **(a)** shows the case without geophysical constraints, while the lower panel shows the case with geophysical constraints. **(b)** **(d)**: Entropy maps based on MCP realizations along the transect at $Y = 6201300$ (UTM). The upper panel shows the case without geophysical constraints, while the lower panel shows the case with geophysical constraints.

**13.**

**Comment: Line 504, is the interpreted geological model (Figure 7) used as reference in the sensitivity analysis? Please clarify.**

**Original:**

[Figure]

**Figure 18.** Sensitivity analysis of Training Image (TI) selection on lithological predictions using the MCP framework with geophysical constraints. The bar plot presents the average Jaccard dissimilarity index computed over 100 realizations for each of the 13 TI scenarios. These include 11 randomly selected TIs, one "Integrated TI" constructed from all 8 transects, and one "Best TI" using the current transect as the TI. Blue bars represent results under a low weight of geophysical constraint, while orange bars represent results under a high weight of geophysical constraint.

**Revision:** Reply to the reviewer: Yes. The sensitivity analysis used the interpreted geological model shown in the bottom right of Figure 7 (transect at Y=6201300 m) as the reference. For each TI selection option, we computed the Jaccard dissimilarity index between each of the 100 MCP realizations and this reference model, and then averaged the results to quantify the dissimilarity between the predicted and reference models for each TI choice.

---

## Author Comment (AC2)

We sincerely appreciate the reviewer for providing these comments that helped us improve the manuscript. Detailed responses are provided below, and the manuscript has been modified accordingly based on each response.

**General comments:**

**1.**

**Comment: The use of airborne electromagnetic (AEM) data is promising, as it can provide extensive subsurface coverage. However, could the authors discuss whether commonly used drone-based electromagnetic systems (which are more accessible and cost-effective in some settings) could provide suitable information for this approach, or are there limitations in resolution or depth penetration that make them incompatible?**

**Reply to the reviewer:** Thanks for the question. Yes, UAV-EM data or towed-TEM systems can substitute SkyTEM data within the MCP framework. This is fundamentally because the MCP framework is inherently scale independent. It operates on categorical bivariate probabilities regardless of the absolute physical dimensions, allowing it to adapt effectively to both shallow, high-resolution investigations and deep, catchment-scale modeling. In general, geophysical datasets with greater depth penetration and higher resolution provide stronger and more informative soft constraints for catchment scale hydrogeology. Regarding drone-based systems, we can distinguish between full UAV-borne systems which carry the transmitter and are typically limited to shallow depths due to lower magnetic moments and semi-airborne systems which use ground-based transmitters to achieve greater depth penetration (Rochlitz et al., 2025). Even if a standard UAV-EM system only provides shallow resistivity information, it can still be highly valuable and integrated into the MCP framework for shallow characterization, provided that an adapted training image is used.

**2.**

**Comment: The authors developed a depth-dependent conditional probability method. As far as I know, depth is location-dependent. Is it possible to use other more easily measured terms, such as bulk density, instead of depth to develop the dependent conditional probability method?**

**Reply to the reviewer:** Yes, the depth of a specific interface is indeed a location-dependent variable. In sedimentary environments, depth often exhibits strong stratigraphic trends, as illustrated in the used training image. Therefore, one geological formation might appear within given depth range. For this reason, depth acts as a meaningful stratigraphic indicator combined with physical property (resistivity) in our conditional probability estimation. Indeed, two geological formations with similar resistivity characteristics can be successfully identified if they have different depth range. This is why we included depth-dependent conditional probability. We will clarify this aspect in the text.

We agree that other measurable properties, such as bulk density, could be used to construct conditional probabilities when such information is available. However, bulk density is typically derived from laboratory measurements and is not routinely or continuously available across

boreholes. In contrast, depth is directly observed in all boreholes and can be consistently combined with AEM resistivity across the entire study area. Therefore, using depth-dependent probability functions represents a practical and geologically meaningful choice in our study, while the incorporation of other physical variables remains a valuable direction for future work. An example of multi-properties conditional probability is discussed in Isunza-Manrique et al. (2023).

**3.**
**Comment: The MCP method is a form of multiple-point statistics based on bivariate transitions. Based on a review of its advantages and disadvantages (e.g., advantages: computationally efficient, enforces stratigraphic ordering via zero-forcing, less reliant on highly repetitive training images; disadvantages: may produce unrealistic outputs in sparse data areas due to fallback on marginal probabilities, assumes conditional independence which can oversimplify complex patterns, and struggles with non-stationarity without extensions), could the authors address how their enhanced approach mitigates general limitations of standard MCP, such as handling non-stationarity or reducing artifacts in data-sparse regions.**

**Reply to the reviewer:** We agree that standard MCP has a few limitations, including the use of marginal probabilities in data-sparse regions, the conditional independence assumption, and its difficulty in representing non-stationary facies trends. Our enhanced MCP framework incorporates geophysical constraints and addresses these limitations in several ways:

1) **Mitigating marginal-probability fallback in sparse-data regions.**
   The integration of AEM derived soft probabilities provides spatially continuous subsurface information, preventing the simulation from reverting to unconditional facies proportions where borehole control is absent. This significantly reduces unrealistic artifacts in areas without any hard constraints.
2) **Reducing the impact of the conditional independence assumption.**
   The standard MCP relies on a conditional independence assumption, where the influence of neighboring facies is modeled only through pairwise bivariate transitions. This simplification limits its ability to capture complex multipoint geological structures. Our enhanced MCP framework mitigates this limitation by integrating geophysically derived conditional probabilities and stratigraphic trends through the permanence-of-ratios formulation, therefore introducing additional conditional dependencies beyond the bivariate transitions used in standard MCP.
3) **Handling vertical non-stationarity.**
   A key property of the MCP framework is zero-forcing behavior. When bivariate probabilities are extracted from the training image, Equation (1) ensures that if any neighboring facies provides a zero probability for a given category, that category will not be simulated at the current location. Therefore, when the training images exhibit strict stratigraphic ordering, MCP realizations naturally reproduce this vertical ordering, except in cases where insufficient neighborhood information leads to fallback on marginal probabilities.

   The conditional probabilities derived from geophysical constraints do not enforce such

strict stratigraphic ordering, but they provide spatially continuous and geologically realistic soft information that reflects the true subsurface complexity. By combining the MCP transition probabilities with geophysical conditional probabilities through the permanence-of-ratios formulation, and by adjusting their relative influence using the parameter τ, the enhanced framework can simultaneously preserve stratigraphic ordering while capturing the vertical heterogeneity expressed by the geophysical data. This allows the method to represent non-stationary geological structures more comprehensively than standard MCP.

4) **Reducing artifacts through structural post-processing.**

The post-processing step targets fine-scale stratigraphic and spatial inconsistencies. Importantly, cells identified as inconsistent are removed (reset to an unsimulated state) and then resimulated within the iterative framework. This allows these locations to be re-estimated based on a corrected neighborhood configuration, ensuring that the final realizations are spatially coherent and aligned with the intended geological structure.

**4.**

**Has the resistivity-lithology relationship mentioned by the authors been validated through independent datasets or field experiments beyond the statistical linking in this study?**

**Reply to the reviewer:** Thanks for pointing this out. In this study, the resistivity–lithology relationship is derived from paired lithological logs and AEM resistivities at the borehole locations and is therefore based on site-specific ground truth. We acknowledge that this relationship has not been validated with independent datasets beyond the borehole informed statistical analysis. However, the resulting trends are consistent with known sedimentological patterns and with resistivity ranges commonly reported for glacial and fluvial deposits in similar settings (Madsen et al., 2022). The goal of the KDE based relationship is not to establish a universal petrophysical law, but to capture the local association between resistivity and facies, which is standard practice in hydrogeophysical joint modeling. The relationship is therefore not meant to be used in another geological setting. This will be added to the discussion part.

**5.**

**Comment: Regarding the statement "More advanced inversion techniques (Deleersnyder et al., 2023) could potentially improve conditional probability from geophysical data as suggested in Hermans and Irving (2017)": Can the authors elaborate on what specific advanced inversion techniques they have in mind (e.g., full-waveform or joint inversions)? Why is the current inversion technique (1D deterministic) not sufficient, and how might these alternatives quantitatively improve the conditional probabilities or reduce smoothing effects?**

**Reply to the reviewer:** A key limitation is that both 1D deterministic inversion and LCI cannot adequately represent two-dimensional or three-dimensional structural variability, which reduces the informativeness of the conditional probability P(lithology| geophysics, depth). Any deterministic inversion using a regularization approach is introducing some prior information that will be mirrored in the inversion result. In case of sharp interface, the smoothness-constraint inversion will lead to smoothing, so that the the conditional probabilities could be less sharp that if a more appropriate blocky inversion would be used for example. Therefore, other geophysical methods could lead to

more discriminating conditional probability as show in Hermans and Irving (2017).

The 'advanced inversion techniques' we refer to are those designed to mitigate this smoothing effect and allow for sharper boundaries. These include blocky inversion methods (e.g., Minimum Gradient Support), joint inversion, or the quasi-2D wavelet-based inversion proposed by Deleersnyder et al. (2023).

Specifically, the method by Deleersnyder et al. (2023) introduces scale-dependent, anisotropic regularization. Unlike standard LCI, it allows vertical boundaries to remain sharp (preserving layer interfaces) while maintaining appropriate lateral continuity. This approach better decouples vertical and horizontal variability, preventing strong lateral changes from creating artificial vertical smearing artifacts. Quantitatively, by recovering sharper and more accurate resistivity contrasts, these methods reduce the variance within lithological classes and minimize the overlap between them. This leads to 'sharper' conditional probability distributions (i.e., higher certainty), thereby improving the reliability of the soft constraints entered into the MCP framework.

**6.**
**Comment: This method seems to integrate soft and hard datasets effectively. However, as physics-informed neural networks (PINN) are becoming popular and the authors mention inversion and other methods, is it possible to integrate the proposed approach into a PINN framework by incorporating both soft (geophysical) and hard (borehole) datasets for potentially more robust predictions?**

**Reply to the reviewer:** Thank you for your suggestion. In our framework, the conditional probability P(lithology| resistivity, depth) derived at borehole locations acts as a soft constraint for MCP, and the primary goal of this component is to improve the quality of the simulations. The potential role of PINNs or other machine learning methods should be to enhance the P(lithology| resistivity, depth) when the geophysical data are noisy or strongly smoothed. However, if improved inversion techniques mentioned before can yield high-quality resistivity models, then a simple probabilistic link provides the most transparent and least biased constraint. In this sense, improving the physical accuracy of geophysical data is more effective in enhancing the soft constraint than using more complex neural network methods to link resistivity with lithology. PINN could also be used in inversion frameworks, for example to directly generate realistic simulations that would fit the geophysical data. This is an interesting area for future research.

**7.**
**Comment: The soft constraints show limited effectiveness in complex structures. As seen in the lower part of Figure 3b, the improvements from soft constraints are limited at stratigraphic interfaces or areas with strong resistivity contrasts, where the interface positions are still primarily controlled by the inversion model. This suggests that the method performs well under conditions where the resistivity-lithology relationship is relatively simple and interfaces are gentle. Does this indicate that the constraining effect may significantly weaken in environments with complex resistivity distributions or overlapping lithologies?**

**Reply to the reviewer:** The reviewer's observation is correct. As shown in the lower part of Figure 3, regions where the inverted resistivity is strongly smoothed, the geophysical information becomes less diagnostic, and therefore the soft constraints naturally have limited influence.

[Figure]

**Figure 3.** (a) True synthetic conductivity model constructed over 50 vertical soundings (columns) using random Gaussian distribution. (b) Both observed and fitted TEM responses for example of Column 9, and comparison between the true and inverted conductivity profiles (bottom). (c) Final stitched 2D conductivity section obtained by combining 1D inversion results for all columns.

This behavior is expected because the resolution of inversion decreases with depth, causing the deeper portion of the recovered model to deviate more strongly from the true conductivity, as shown in the 3rd scenario of results from the synthetic case. Even though the conditional probability derived from geophysics becomes more uncertain at depth or complex area, due to overlapping lithologies or reduced inversion resolution, the MCP framework does not rely solely on the geophysical information at these locations. In regions where the resistivity–lithology relationship is distinct (e.g., shallower areas or zones close to boreholes), it produces well-constrained simulated facies. These well-resolved facies act as strong neighborhood conditioning data within the MCP algorithm and propagate information toward the deeper or more ambiguous regions.

This mechanism is clearly illustrated in the result of the synthetic test (Figure 4): despite the poor recovery of deeper conductivity in the inversion, introducing only a single column of borehole lithology data was sufficient to correct the complex structures at depth. The lower part of the model, which appeared highly uncertain when conditioned only on geophysical probabilities, became significantly more accurate once the MCP propagation incorporated the borehole information. This behavior reflects the core strength of the proposed integration that the conditional probability from geophysics provides broad-scale spatial patterns, while MCP and borehole data enforce local consistency and stratigraphic structure. The two constraints compensate for each other's influence, allowing the framework to remain robust even where geophysical soft constraints alone become overlap or hard data is not sufficient.

**8.**

**Comment: Kernel density estimation (KDE) is a core step in establishing the lithology-resistivity relationship (Section 3.2), but the manuscript does not specify the method for determining the bandwidth. It is recommended to clearly state the basis for selecting the bandwidth and discuss its sensitivity to ensure the reproducibility of the results.**

**Reply to the reviewer:** Thank you for raising this point regarding the transparency of the KDE parameters. The bandwidth in our KDE-based estimation of conditional probabilities is determined automatically using MATLAB's multivariate kernel density estimator with the normal reference rule ('Bandwidth', 'normal').

This rule is a standard, data-adaptive bandwidth selection method that optimally computes the bandwidth based on the sample covariance and the number of data points in each lithology class. It minimizes the mean integrated squared error for Gaussian-like distributions. Regarding sensitivity, this data-driven approach provides a robust balance between bias (over-smoothing) and variance (under-smoothing), ensuring that the resulting probability distributions are stable and not sensitive to subjective manual tuning.

**Original:** *'Kernel Density Estimation (KDE) is employed in place of simple histogram based sampling (Manrique et al., 2024). By modeling the joint probability distribution in a continuous probabilistic framework, KDE not only helps smooth out local variability but is particularly well-suited to this scenario, where depths are only available at sparse borehole locations and resistivity values are derived from constrained 1D inversions, thereby yielding more reliable conditional probabilities in data-sparse zones.'*

**Revision:** '*Kernel Density Estimation (KDE) is employed in place of simple histogram based sampling (Manrique et al., 2024). By modeling the joint probability distribution in a continuous probabilistic framework, KDE not only helps smooth out local variability but is particularly well-suited to this scenario, where depths are only available at sparse borehole locations and resistivity values are derived from constrained 1D inversions, thereby yielding more reliable conditional probabilities in data-sparse zones. The bandwidth in our KDE based estimation of conditional probabilities is determined using a standard, data adaptive bandwidth selection method that computes the bandwidth from the sample covariance and the number of data points in each lithology class*'

**9.**

**Comment: The regularization parameters for the 1D TEM inversion are missing. The inversion uses GNCG + Tikhonov regularization (Section 3.1), but key parameters are not provided, including: the value of the regularization coefficient λ and its determination method; and the strategy for balancing data fit and model smoothness. This information is crucial for assessing the reliability of the inversion and reproducing the method.**

**Reply to the reviewer:** Thanks for pointing out the need to specify the regularization parameters of the 1D TEM inversion. We have now added a complete description of the misfit, regularization,

and trade-off terms in Section 3.1. The inversion uses a weighted L2 data misfit with 2% relative uncertainties and a Tikhonov model objective function that includes both zero-order (smallness) and first-order vertical smoothness terms with $\alpha_s = 0.01$ and $\alpha_x = 1.0$.

The regularization coefficient $\lambda$ is automatically estimated from the largest eigenvalue of the approximate Hessian (using a ratio of 10), following a standard eigenvalue-based strategy. The resulting $\lambda$ is kept fixed during the IRLS updates, providing a stable balance between data fit and smoothness.

The optimization is carried out using a projected GNCG algorithm with specified iteration limits and convergence tolerances. These details ensure that the inversion is fully reproducible, and they have now been included in the 3.1.2 of the revised manuscript.

**10.**

**Comment: Figure 5 shows that $\tau \approx 3$ is optimal in the synthetic case, but the final $\tau$ value used in the Egebjerg field case is not specified. It is suggested to clearly state the value of $\tau$ at the beginning of Section 3.2 and explain the basis for its selection (e.g., sensitivity analysis, data quality, or borehole density).**

**Reply to the reviewer:** Thank you for raising this important point. We would like to emphasize that $\tau$ in the permanence-of-ratios formulation is not intended to be a fixed or universal value, instead, it acts as an adaptive weighting parameter whose optimal value depends on the relative reliability of the geological (MCP based) and geophysical components at a given site.

In the synthetic case, the Training Image (TI) was fully representative, and borehole data were available along the entire vertical extent. Under these idealized conditions, sharpening the geophysical contribution through $\tau = 3$ produced the best results, as demonstrated in the sensitivity analysis (Figure 5). The sensitivity analysis in the synthetic case also demonstrates that there exists an intermediate value of $\tau$ that optimally balances the geological and geophysical contributions. However, the specific optimal value is site-dependent, as it must reflect the relative reliability and trustworthiness of the geological prior and the geophysical information at each location.

In contrast, the Egebjerg field data present a fundamentally different configuration: almost all boreholes are shallow (<50 m), meaning that MCP-based probabilities revert to marginal proportions at depth and become unreliable for representing deep structures such as buried valleys. If we enhanced the geophysical term (c) in the same way as in the synthetic case, the unreliable geological prior (b) would still compete with the geophysical information and introduce inconsistencies at depth. For this reason, we adopted a site-specific, adaptive weighting strategy. Instead of sharpening c, we applied a dampening factor $\tau = 0.11$ to the MCP prior term (b). This reduces the influence of depth-varying but unreliable MCP probabilities in poorly informed regions, allowing the geophysical constraints to dominate in zones where borehole data do not constrain deeper structures.

This adaptive use of $\tau$ is consistent with the intended flexibility of the permanence-of-ratios

formulation and ensures that the combined probability reflects the relative reliability of each component. We have now explicitly added the value τ = 0.11 and the rationale for its selection at the beginning of Section 3.2.4 in the revised manuscript.

**Original**:'*The final step is to integrate both geological expert knowledge represented in the TI and geophysical information using the permanence of ratios principle Equation (5) within the MCP framework, generating geologically realistic lithological realizations constrained by both borehole and geophysical data.*'

**Revision:** '*The final step is to integrate both geological expert knowledge represented in the TI and geophysical information using the permanence of ratios principle (Equation 5) within the MCP framework. In this specific application, an adaptive weighting strategy is employed by applying a dampening parameter of τ=0.11 to the MCP prior term (b). This adjustment mitigates the influence of unreliable MCP information in deep zones because of shallow boreholes (< 50m), generating geologically realistic lithological realizations constrained by both borehole and geophysical data.*'

**11.**
**Comment: In the Code Availability section, the authors state that the simulation codes are proprietary and available upon reasonable request. To improve reproducibility, transparency, and accessibility for the scientific community, it is recommended that the codes be made publicly available, for example, on a platform like GitHub or Zenodo, if possible.**

**Reply to the reviewer:** We fully agree with the reviewer that open science and code accessibility are vital for reproducibility and transparency in the scientific community. We would very much like to share the code if it were within our authority. However, the core MCP simulation algorithm used in this study is the proprietary intellectual property of the Geological Survey of Canada. As users of this code under a specific collaboration agreement, we are restricted from distributing the source code or hosting it on public platforms like GitHub or Zenodo. As stated in the 'Code Availability' section, we can facilitate access to the code for non-commercial research purposes upon email request. For more details of the MCP description, could refer to the paper (Benoit, N., Marcotte, D., Boucher, A., D'Or, D., Bajc, A., and Rezaee, H.: Directional hydrostratigraphic units simulation using MCP algorithm, Stochastic Environmental Research and Risk Assessment, pp. 1435–1455, 2018.)

**Specific comments:**

**1.**
**Comment: Line 48: The dot before "(He et al., 2014)" should be removed.**

**Original**:'*… probabilistic constraints (e.g. resistivity models derived from inverted airborne geophysical measurements), and prior geological knowledge about spatial variability and continuity (He et al., 2014; Høyer et al., 2017; Barfod et al., 2018).*'

**Revision:** '*… probabilistic constraints (e.g. resistivity models derived from inverted airborne*

*geophysical measurements), and prior geological knowledge about spatial variability and continuity (He et al., 2014; Høyer et al., 2017; Barfod et al., 2018).'*

**2.**
**Comment: Line 231: Please remove the extra "a" in the sentence "the a".**

**Original:** *'5. The bivariate probabilities are derived from the synthetic training image, which are then used to calculate the conditional probabilities for the MCP function as in Equation (1). In the constrained scenarios, a single borehole located X=8m is introduced as hard conditioning data.'*

**Revision:** *'5. The bivariate probabilities are derived from the synthetic training image, which are then used to calculate the conditional probabilities for the MCP function as in Equation (1). In the constrained scenarios, a single borehole located X=8m is introduced as hard conditioning data.'*

**3.**
**Comment: Line 263: There is a repetition of "Lithology = 12"; please correct this.**

**Reply to the reviewer:** We assume the reviewer refers to Line 363, where we discuss the bivariate probabilities for the co-existence of lithology 12 at different locations. We have modified the sentence to make it clearer.

**Original :** '*Figure 8 shows an example of the integrated bivariate probability distribution P(Lithology = 12, Lithology = 12).*'

**Revision:** '*Figure 8 shows an example of the integrated bivariate probability distribution P(i,j) at different locations, where both i and j correspond to lithology 12.*'

**4.**
**Comment: Line 352: "We" in "We mimic" should be lowercase (change to "we" for consistency if it's not starting a sentence, or check context).**

**Original:** '*Since a complete 3D TI is often unavailable in practice, We mimic this limitation by extracting 2D transects...*'

**Revision:** '*Since a complete 3D TI is often unavailable in practice, we mimic this limitation by extracting 2D transects...*'

**5.**
**Comment: Line 402: For "a few deep boreholes," the authors should specify the depth range that qualifies as "deep" boreholes, how many boreholes fall into this category versus shallow ones, and the threshold used to distinguish them.**

**Reply to the reviewer:** Thanks for the comment. In our study, as shown in Figure 6 and stated in Section 3.2.1, the study area contains 794 boreholes, but they are predominantly shallow:

approximately 78% of them terminate at depths of less than 50 m (Ter-Borch, 1991). In contrast, our hydrostratigraphic model extends vertically from +80 m to -100 m elevation, requiring characterization of structures significantly deeper than the reach of most boreholes.

Furthermore, for the two specific transects selected for detailed analysis, the borehole coverage is even more limited, with many boreholes terminating at depths of less than 20 m. This sharp contrast between the shallow hard data and the deep target structures (e.g., buried valleys) is the primary motivation for integrating deep penetrating AEM data.

**6.**
**Comment: Line 422-425: I suggest adding explicit references to Figure 12a and 12b to more clearly explain the related sub-figures.**

**Original:** '*In the absence of geophysical constraints and with limited hard data (borehole) support, the MCP simulations tend to produce more simplified and uniform stratigraphic configurations....*'

**Revision:** *'As shown in Figures 12(a–b) and 13(a–b), in the absence of geophysical constraints and with limited hard data (borehole) support, the MCP simulations tend to produce more simplified and uniform stratigraphic configurations...'*

**7.**
**Comment: Line 435: It should also clearly mention which subfigure of Figure 7 is being referred to.**

**Original:** '*In the absence of geophysical constraints and with limited hard data (borehole) support, the MCP simulations tend to produce more simplified and uniform stratigraphic configurations....*'

**Revision:** *'As shown in Figures 12(a–b) and 13(a–b), in the absence of geophysical constraints and with limited hard data (borehole) support, the MCP simulations tend to produce more simplified and uniform stratigraphic configurations...'*

**8.**
**Comment: Line 458: The entropy maps are not clear. Please provide more details regarding why entropy is used (e.g., as a measure of predictive uncertainty) and a more detailed interpretation of this figure, including how variations in entropy relate to data constraints or geological features.**

**Reply to the reviewer:** Thanks for the comment about entropy plots, we have added more details related to this into the manuscript. The entropy maps (Right panels of Figures 14 and 15) illustrate the spatial distribution of predictive uncertainty, where Shannon entropy serves as a quantitative metric of randomness in the lithological predictions. High entropy values (black zones) indicate areas where the model predictions are variable across realizations, while low entropy values (red/yellow zones) reflect high confidence. Shannon entropy is widely established in geostatistical

literature as a robust indicator of categorical uncertainty (e.g., Lindsay et al., 2021; Madsen et al., 2021; Pirot et al., 2022).

Geologically, variations in entropy correlate with structural complexity. Uniform and continuous layers (e.g., deep clay layers) generally exhibit lower entropy. However, higher entropy is observed along the boundaries of buried valleys, reflecting the high variability in precisely resolving abrupt lithological transitions and complex erosional features in the absence of direct hard data.

The presence of these localized high-entropy corridors is therefore an indicator that the model has successfully identified the buried-valley geometry. With geophysical constraints, the entropy is spatially structured, low in the continuous clay units and higher only along the valley flanks, revealing where information is well constrained and where ambiguity remains. In contrast, without geophysical constraints, entropy remains high across much of the model domain, and the buried valley structure cannot be resolved at all. Thus, the entropy maps complement the lithology realizations by showing not only that the soft constraints help recover the valley architecture, but also how uncertainty is distributed in a geologically interpretable way.

**9.**
**Comment: The paper extracts 88 2D transects from a 3D model to compute transition probabilities (around Line 353), but does not justify whether the sample is sufficient to characterize the spatial variability of the study area. It is recommended to add a brief statistical description of the transect orientations, spacings, and coverage uniformity to strengthen the argument for representativeness.**

**Reply to the reviewer:** We thank the reviewer for pointing this out. We agree that justifying the sampling density is important for representativeness.

In our study, the 88 transects were extracted using a systematic grid sampling strategy, with a uniform spacing of 400 m along both orthogonal X (546600 m - 564300 m) and Y (6188700 m - 6205700 m) directions. This dense coverage ensures that the spatial variability of the hydrostratigraphic model is fully captured across the entire domain.

To further confirm this sufficiency, we refer to the sensitivity analysis in Section 4 (Figure 19), where we tested the model using only 11 randomly selected transects. The results showed that the simulations are robust and insensitive to the reduction in TI sample size, confirming that the full set of 88 transects is more than sufficient. We have added these details to Section 3.2.2 to strengthen the argument.

**10.**
**Comment: Figure 2: Inconsistent coordinate ranges and units. The horizontal coordinate ranges differ between the left and right subfigures, and the colorbar units are missing (presumed to be S/m). It is suggested to unify the coordinate ranges and label the units.**

**Reply to the reviewer:** Thank you for the notice, we have redrawn Figure 2 according to your

suggestions.

**Original:**

[Figure]

**Figure 2.** Left: Synthetic lithological model. Right: Synthetic conductivity distribution generated from the Gaussian random field.

**Revision:**

[Figure]

**Figure 2.** a): Synthetic lithological model. b): Synthetic conductivity distribution generated from the Gaussian random field.

**11.**

**Comment: Figure 8: Inconsistent horizontal coordinate ranges. The left figure spans -89 to 87, while the right spans -89 to 89. It is suggested to unify the coordinate ranges for easier comparison.**

**Reply to the reviewer:** We thank the reviewer for the notice. We have redrawn Figure 8 to enforce consistent horizontal coordinate ranges. Both the left and right subplots now span the unified range of [-89, 89] to facilitate direct comparison.

**Original:**

[Figure]

**Figure 8.** Bivariate transition probabilities $P(i, j)$, where $i$ and $j$ represent the 14 lithological layers, were derived from 88 transects extracted from the 3D interpreted model. These were then merged into a single generalized 2D bivariate probability distribution using the Extended Logistic Opinion Pool (ELOP) method. This unified distribution can be applied across all 2D transects. Shown here is an example for $P(\text{Lithology} = 12, 12)$ illustrating the spatial relationship aggregated from all 88 transects.

**Revision:**

[Figure]

**Figure 8.** Bivariate transition probabilities $P(i, j)$, where $i$ and $j$ represent the 14 lithological layers, were derived from 88 transects extracted from the 3D interpreted model. These were then merged into a single generalized 2D bivariate probability distribution using the Extended Logistic Opinion Pool (ELOP) method. This unified distribution can be applied across all 2D transects. Shown here is an example for $P(\text{Lithology} = 12, 12)$ illustrating the spatial relationship aggregated from all 88 transects.

**12.**

**Comment: Inconsistent scaling. The right figure includes a "×10⁵" label, while the left does not. It is suggested to unify the scaling.**

**Reply to the reviewer:** Thank you for the notice of missing scaling label, we have redrawn Figure 9 according to your suggestions.

**Original:**

[Figure]

Figure 9. (a): Lithological information of each borehole. (b): Inverted resistivity values are interpolated along the borehole locations. (c): Inverted resistivity distribution with each lithology type based on interpolated resistivity values.

**Revision:**

[Figure]

Figure 9. (a): Lithological information of each borehole. (b): Inverted resistivity values are interpolated along the borehole locations. (c): Inverted resistivity distribution with each lithology type based on interpolated resistivity values.

**13.**

**Comment: Mixed use of conductivity/resistivity units. Both S/m and Ω·m appear in the text and figures. It is suggested to standardize to Ω·m and explain the conversion relationship upon first appearance.**

**Reply to the reviewer:** We thank the reviewer for the suggestion regarding unit consistency. However, for the synthetic case, we have chosen to retain the units of conductivity (S/m). Since the regularization and sensitivity scaling in the inversion scheme were based on conductivity, it faithfully represents the actual parameterization used in our forward modeling and inversion algorithms (Figure 3). Converting these values to resistivity in the description would misrepresent the methodological setup. To bridge the gap with the field case (which uses resistivity) and assist the reader, we have added a clarification of the conversion relationship at the first mention of the unit of resistivity in Section 3.2.3.

**Original:** '*Unlike the synthetic case, where a 1D deterministic inversion was used, the TEM data were inverted using the Laterally Constrained Inversion (LCI) method (Auken et al., 2005), which applies spatial constraints to ensure lateral continuity between adjacent soundings, resulting in more geologically consistent resistivity profiles (Madsen et al., 2022). This is especially important in areas with limited borehole control, as it reduces inversion artifacts and noise (Høyer et al., 2015; Jørgensen et al., 2015).*'

**Revision:** '*Unlike the synthetic case, where a 1D deterministic inversion was used, the TEM data were inverted using the Laterally Constrained Inversion (LCI) method (Auken et al., 2005), which applies spatial constraints to ensure lateral continuity between adjacent soundings, resulting in more geologically consistent resistivity profiles (Madsen et al., 2022). This is especially important in areas with limited borehole control, as it reduces inversion artifacts and noise (Høyer et al., 2015; Jørgensen et al., 2015). In the real-case study, the resistivity data (Ω·m) are used to statistically link lithology. Resistivity has a direct conversion relationship with conductivity used in the synthetic case (resistivity = 1 / conductivity).*'

**14.**
**Comment: Figure 10: Abnormal format in the Layer 14 subfigure. The subplot for Layer 14 has inconsistent sizing and blank areas. It is suggested to revise it.**

**Reply to the reviewer:** Thank you for the notice, we have modified Figure 14 accordingly.

**Original:**

[Figure]

**Figure 10.** The heatmaps present the smoothed probabilistic distributions of each lithological layer estimated via KDE, conditioned on resistivity and depth. The red lineplot in each subplot represents the global normalized marginal probability of lithology conditioned solely on resistivity, derived from the KDE results. The boxplots below each heatmap display the distribution of resistivity values at each layer based on inverted geophysical data at borehole locations.

**Revision:**

[Figure]

**Figure 11.** The heatmaps present the smoothed probabilistic distributions of each lithological layer estimated via KDE, conditioned on resistivity and depth. The red lineplot in each subplot represents the global normalized marginal probability of lithology conditioned solely on resistivity, derived from the KDE results. The boxplots below each heatmap display the distribution of resistivity values at each layer based on inverted geophysical data at borehole locations.

**References:**

Benoit, N., Marcotte, D., Boucher, A., D'Or, D., Bajc, A., and Rezaee, H.: Directional hydrostratigraphic units simulation using MCP algorithm, Stochastic Environmental Research and Risk Assessment, 1435–1455, 2018.

Deleersnyder, W., Maveau, B., Hermans, T., and Dudal, D.: Flexible quasi-2D inversion of time-domain AEM data, using a wavelet-based complexity measure, Geophysical Journal International, 233, 1847–1862, 2023.

Hermans, T. and Irving, J.: Facies discrimination with electrical resistivity tomography using a probabilistic methodology: effect of sensitivity and regularisation, Near Surface Geophysics, 15, 13–25, 2017.

Lindsay, M. D., Aillères, L., Jessell, M. W., de Kemp, E. A., and Betts, P. G.: Locating and quantifying geological uncertainty in three-dimensional models: Analysis of the Gippsland Basin, southeastern Australia, Tectonophysics, 546, 10–27, 2012.

Madsen, R. B., Møller, I., and Hansen, T. M.: Choosing Between Gaussian and MPS Simulation: The Role of Data Information Content—A Case Study Using Uncertain Interpretation Data Points, Stochastic Environmental Research and Risk Assessment, 35, 1563–1583, 2021.

Madsen, R. B., Høyer, A.-S., Andersen, L. T., Møller, I., and Hansen, T. M.: Geology-driven modeling: A new probabilistic approach for incorporating uncertain geological interpretations in 3D geological modeling, Engineering Geology, 309, 106 833, 2022.

Manrique, I. I., Caterina, D., Nguyen, F., and Hermans, T.: Quantitative interpretation of geoelectric inverted data with a robust probabilistic approach, Geophysics, 88, B73–B88, 2023.

Pirot, G., Joshi, R., Giraud, J., Lindsay, M. D., and Jessell, M. W.: loopUI-0.1: indicators to support needs and practices in 3D geological modelling uncertainty quantification, Geoscientific Model Development, 15, 4689–4708, 2022.

Rochlitz, R., Günther, T., Kotowski, P. O., and Becken, M.: Semi-airborne electromagnetic exploration of deep sulfide deposits with UAV-towed magnetometers — Part 2: Inversion and resolution analysis, Geophysics, 90, WA307–WA322, 2025.